# BRCA2 stabilises RAD51 and DMC1 nucleoprotein filaments through a conserved interaction mode

James M. Dunce[1] & Owen R. Davies [2] ✉

BRCA2 is essential for DNA repair by homologous recombination in mitosis and meiosis. It interacts with recombinases RAD51 and DMC1 to facilitate the formation of nucleoprotein filaments on resected DNA ends that catalyse recombination-mediated repair. BRCA2's BRC repeats bind and disrupt RAD51 and DMC1 filaments, whereas its PhePP motifs bind recombinases and stabilise their nucleoprotein filaments. However, the mechanism of filament stabilisation has hitherto remained unknown. Here, we report the crystal structure of a BRCA2-DMC1 complex, revealing how core interaction sites of PhePP motifs bind to recombinases. The interaction mode is conserved for RAD51 and DMC1, which selectively bind to BRCA2's two distinct PhePP motifs via subtly divergent binding pockets. PhePP motif sequences surrounding their core interaction sites protect nucleoprotein filaments from BRC-mediated disruption. Hence, we report the structural basis of how BRCA2's PhePP motifs stabilise RAD51 and DMC1 nucleoprotein filaments for their essential roles in mitotic and meiotic recombination.

DNA double-strand break (DSB) repair by homologous recombination is critical for genome integrity and fertility[1–3]. In somatic cells, DSBs arise due to exogenous damage and upon replication fork collapse[2]. These lesions can be repaired, and replication can be restarted through inter-sister recombination. The machinery of inter-sister recombination also has an additional role in protecting stalled replication forks[4,5]. In meiosis, a programme of DSB induction triggers inter-homologue recombination[1], enabling synapsis between homologues[6], and the formation of crossovers that ensure correct chromosome segregation and enhance genetic diversity[7,8]. Hence, defects in homologous recombination are associated with chromosome instability, increased cancer risk and infertility[9–11].

The tumour suppressor BRCA2 performs a central role in the mechanics of recombination by loading recombinases onto resected DNA ends to form nucleoprotein filaments that mediate strand invasion and homology search within the template DNA[12–14]. This involves the displacement of RPA from newly resected DNA ends[15,16], and the remodelling of recombinases from their oligomeric assemblies into protein-DNA filaments that are active in recombination[12–14]. Whilst

RAD51 is the universal recombinase, meiosis also requires DMC1[17,18], which enables mismatch-tolerant inter-homologue recombination[19]. BRCA2 and RAD51 knockouts are embryonically lethal[20,21], their disruption in cell lines leads to gross chromosomal rearrangements[10,22,23], and human BRCA2 mutations are strongly associated with early-onset breast and ovarian cancers[9]. Further, DMC1 disruption, and germline-specific deficiencies of BRCA2 and RAD51, lead to meiotic impairment and infertility[11,17,18,24].

BRCA2 interacts with RAD51 and DMC1 recombinases through BRC repeats and PhePP motifs (Fig. 1a)[21,25–28]. The eight BRC repeats, located within BRCA2's central exon 11 region, bind to RAD51 and DMC1 with various affinities[25,29,30]. They interact via molecular mimicry with the self-association interface that mediates the oligomeric assembly and nucleoprotein filament formation of recombinases, thereby locking them in inactive monomeric states[31]. Hence, BRC repeats are thought to remodel RAD51 and DMC1, from their isolated states in heterogeneous oligomers and octameric rings, respectively, to nucleate their assembly on resected DNA ends into active nucleoprotein filaments. BRCA2 has two PhePP motifs, located within exons 14 and 27 (herein referred to as

[1]Department of Biochemistry, University of Cambridge, Cambridge, UK. [2]Wellcome Centre for Cell Biology, Institute of Cell Biology, University of Edinburgh, Michael Swann Building, Max Born Crescent, Edinburgh, UK. ✉e-mail: owen.davies@ed.ac.uk

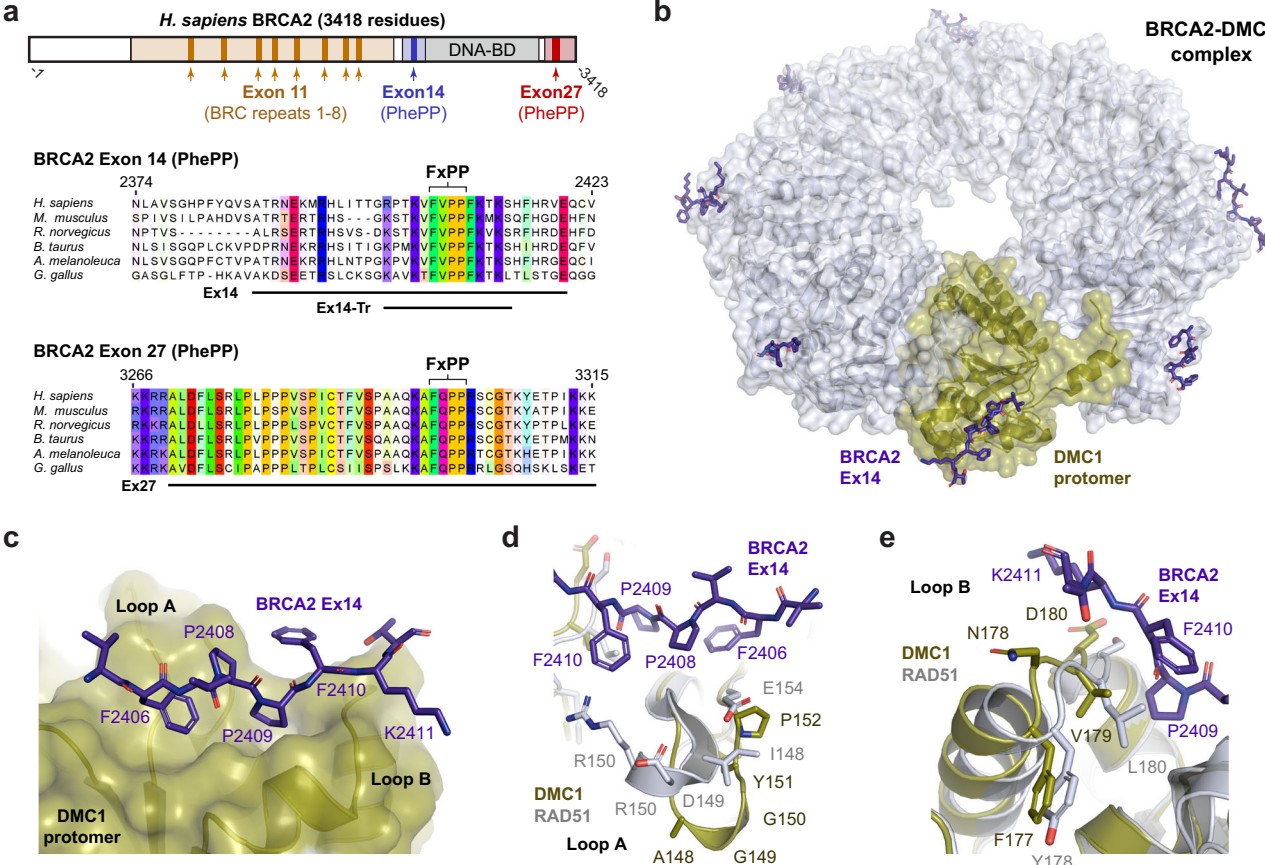

**Fig. 1 | Crystal structure of a BRCA2-DMC1 complex. a** Schematic of the human BRCA2 sequence (top) with multiple sequence alignments of the exon 14 (middle) and exon 27 (bottom) regions, highlighting their PhePP (FxPP) motifs. The constructs used in this manuscript, corresponding to Ex14 (amino-acids 2387-2420), Ex14-Tr (amino-acids 2401-2414) and Ex27 (amino-acids 3270-3315), are indicated. **b** Crystal structure of the complex between BRCA2 Ex14-Tr peptides and a DMC1 ΔN octameric ring. **c** Interaction between the PhePP region of the BRCA2 Ex14 peptide (purple) and a DMC1 protomer (yellow), highlighting FxPP amino-acids F2406, P2408 and P2409, which are bound in a pocket between two loops on the DMC1 surface (labelled as loops A and B). **d, e** Superposition of the BRCA2-DMC1 structure (purple and yellow) with the RAD51 core (white). **d** PhePP-binding loop A interacts with the N-terminal end of the BRCA2 Ex14 peptide, and diverges between recombinases, consisting of 147-GAGGYPG-153 in DMC1 and 148-IDRGGGE-154 in RAD51. **e** PhePP-binding loop B interacts with the C-terminal end of the BRCA2 Ex14 peptide, and also diverges between recombinases, consisting of 177-FNVD-180 in DMC1 and 178-YGLS-181 in RAD51.

Ex14 and Ex27, respectively; Fig. 1a)[21,27,32]. Whilst Ex14 and Ex27 share an eponymous FxPP motif, their sequences are otherwise divergent (Fig. 1a), and they demonstrate specificity for DMC1- and RAD51-binding, respectively[21,27,32]. Importantly, Ex27 binds to RAD51 filaments, rather than monomers, and protects nucleoprotein filaments from BRC-mediated disruption[33,34]. However, it remains unknown how PhePP motifs interact with recombinases.

Here, we report the crystal structure of BRCA2 Ex14 bound to DMC1, and that BRCA2 Ex27 binds to RAD51 through the same core interaction mode. Hence, we uncover the structural basis of how BRCA2 PhePP motifs bind to RAD51 and DMC1 recombinases and stabilise their nucleoprotein filaments for mitotic and meiotic recombination.

## Results
### Crystal structure of a BRCA2-DMC1 complex
The mode of interaction between BRCA2's PhePP motifs and recombinases has remained unknown since the discovery of this filament-stabilising binding site over two decades ago[21,27,33,34]. As DMC1 exists in octameric rings, whereas RAD51 forms filaments, we reasoned that DMC1 may be more amenable for structure solution of a BRCA2 PhePP-bound complex. Hence, we established a crystallisation system for the structural core of human DMC1 in which its N-terminal domain was deleted (herein

referred to as DMC1 ΔN; amino-acids 83–340). We first validated this system by determining the native structure to a resolution of 2.05 Å, using a previous DMC1 structure (PDB accession 4HYY)[35] as a template for molecular replacement (Table 1). We then soaked DMC1 ΔN crystals with a peptide corresponding to a truncated form of BRCA2's Ex14 (herein referred to as Ex14-Tr; amino-acids 2401-2414). Whilst soaking substantially reduced crystal quality, we obtained X-ray diffraction data to anisotropic resolution limits of 3.40–5.80 Å, in which additional peptide density was clearly observed (Fig. 1b, Table 1 and Supplementary Fig. 1). The peptide density was located adjacent to a loop formed by DMC1 amino-acids 178-183, which is a crystal lattice contact. Hence, the reduction in resolution upon BRCA2 Ex14-binding is owing to the introduction of a defect in the crystal lattice.

The electron density maps of the bound BRCA2 Ex14 peptide could not be interpreted unambiguously (Supplementary Fig. 1). Nevertheless, we were able to build the BRCA2 Ex14 structure guided by an *AlphaFold2* model of the same BRCA2 Ex14 sequence bound to a DMC1 ΔN dimer (Supplementary Fig. 2a–e). This resulted in a refined model of the peptide at seven out of the eight DMC1 protomers, with more extensive structural information at instances away from rather than at crystal lattice contacts (Table 1 and Fig. 1b). The observed BRCA2 Ex14 peptide contains the PhePP motif (amino-acids 2406-2409), in which F2406 is buried in a hydrophobic pocket and the two consecutive proline

## Table 1 | Data collection, phasing and refinement statistics

| | DMC1 | BRCA2-DMC1 |
|---|---|---|
| PDB accession | 6R3P | 8R2G |
| **Data collection** | | |
| Space group | I422 | P4$_1$22 |
| Cell dimensions | | |
| a, b, c (Å) | 174.49, 174.49, 179.57 | 125.40, 125.40, 364.195 |
| α, β, γ (°) | 90, 90, 90 | 90, 90, 90 |
| Resolution (Å) | 47.49–2.05 (2.09–2.05)[a] | 182.098–3.452 (3.692–3.452)[a] |
| Ellipsoidal resolution (Å) (direction) | N/A | 3.395 (a[a]) 3.395 (b[a]) 5.793 (c[a]) |
| $R_{meas}$ | 0.096 (2.038) | 0.247 (1.500) |
| $R_{pim}$ | 0.034 (0.720) | 0.072 (0.561) |
| $I/\sigma(I)$ | 19.5 (1.6) | 8.0 (1.5) |
| CC$_{1/2}$ | 1.000 (0.628) | 0.998 (0.386) |
| Completeness (spherical) (%) | 100.0 (100.0) | 57.7 (16.1) |
| Completeness (ellipsoidal) (%) | N/A | 93.1 (69.2) |
| Redundancy | 15.0 (15.5) | 12.2 (7.1) |
| **Refinement** | | |
| Resolution (Å) | 47.01–2.05 | 118.6–3.452 |
| No. reflections | 86291 | 22652 |
| $R_{work}/R_{free}$ | 0.1942/0.2191 | 0.2633/0.3065 |
| No. atoms | 8118 | 15352 |
| Protein | 7736 | 15352 |
| Ligand/ion | 89 | 0 |
| Water | 293 | 0 |
| B-factors | 58.74 | 94.23 |
| Protein | 58.51 | 94.23 |
| Ligand/ion | 90.00 | N/A |
| Water | 55.27 | N/A |
| R.m.s. deviations | | |
| Bond lengths (Å) | 0.002 | 0.002 |
| Bond angles (°) | 0.478 | 0.480 |

[a]Values in parentheses are for highest-resolution shell.

residues enable an unusual backbone conformation (Fig. 1c). The conformation of the Ex14 peptide was the same in all seven copies (r.m.s. deviation <0.4 Å), albeit with variation in the orientation of the F2410 side-chain and the C-terminus (Supplementary Fig. 3a). Further, the conformation was largely the same as the initial *AlphaFold2* model (r.m.s. deviation <0.7 Å), with notable differences observed in C-terminal amino-acids F2410, K2411 and T2412 (Supplementary Fig. 3b).

The PhePP-binding site of DMC1 is formed by two loops, corresponding to amino-acids 147-153 and 178-183, respectively. The first loop consists of amino-acids 147-GAGGYPG-153 in DMC1 and 148-IDRGGGE-154 in RAD51 (herein referred to as PhePP-binding loop A; Fig. 1d), and is one of the most divergent regions between DMC1 and RAD51 (Supplementary Fig. 4). The second loop consists of amino-acids 178-NVDHDA-183 in DMC1 and 179-GLSGSD-184 in RAD51 (herein referred to as PhePP-binding loop B; Fig. 1e), in which residues V179 (DMC1) and L180 (RAD51) substantially contribute to the PhePP-binding pocket. Hence, the sequence differences in the two loops that contribute to the PhePP-binding pocket may account for the binding specificity in which Ex14 and Ex27 bind selectively to DMC1 and RAD51, respectively[21,27,32].

## BRCA2's Ex14 PhePP motif mediates selective DMC1-binding

We next validated the BRCA2-DMC1 structure through interaction studies. Microscale thermophoresis (MST) confirmed that DMC1 ΔN binds to Ex14 (amino-acids 2387-2420) with a $K_D$ of approximately 30 μM, which is over threefold higher affinity than Ex27 (Fig. 2a and Supplementary Fig. 5). Further, the binding affinity was largely retained for the truncated Ex14-Tr peptide used in crystallographic studies (Fig. 2a and Supplementary Fig. 5). In contrast, the interaction was largely disrupted upon introduction of alanine mutations of the PhePP motif (herein referred to as Ex14-AAA; amino-acids F2406A, P2408A, P2409A) (Fig. 2a). These MST results agree with biochemical pull-down assays, in which DMC1 ΔN was shown to bind to Ex14, Ex14-Tr and BRC4, but not Ex14-AAA or Ex27 (Fig. 2b).

The PhePP-binding site is formed by two loops that diverge in sequence between DMC1 and RAD51. Hence, we wondered whether we could alter DMC1's binding specificity by swapping its PhePP-binding loops with the sequences present in RAD51. Thus, we engineered a DMC1 ΔN loop mutant in which PhePP-binding loops A and B were mutated from 147-GAGGYPG-153 and 178-NVDHDA-183 to RAD51 sequences IDRGGGE and GLSGSD, respectively. Importantly, we confirmed that the DMC1 ΔN loop mutant retained its octameric ring structure in solution (Supplementary Fig. 6a–f and Supplementary Table 1). The DMC1 ΔN loop mutant retained its interaction with BRC4, and abolished its ability to bind Ex14, both by MST and pull-down (Fig. 2a, c and Supplementary Fig. 5). However, it also failed to bind Ex27 (Fig. 2c). Hence, replacing the PhePP-binding loops of DMC1 with the RAD51 sequences was insufficient to confer Ex27-binding, indicating that additional sequence and/or structural differences between DMC1 and RAD51 contribute to PhePP-binding specificity. Nevertheless, loss of Ex14-binding confirmed that we had correctly identified the PhePP-binding site. DMC1's PhePP-binding pocket includes V179 (L180 in RAD51) that mediates hydrophobic interactions with Ex14 amino acids P2409 and F2410. Hence, we introduced a point mutation V179E to disrupt this interface. The mutation did not affect DMC1's octameric structure in solution (Supplementary Fig. 6a), and whilst DMC1 ΔN V179E retained binding to BRC4, it failed to interact with Ex14 (Fig. 2d). Hence, the loop and V179E mutants confirm that the PhePP-binding site observed in the crystal structure is responsible for Ex14-binding in solution.

It was previously demonstrated that BRCA2 Ex27-binding is specific to RAD51 oligomers by using an F86E mutation that disrupts RAD51's self-association interface[31,33,34]. Hence, to test whether BRCA2 Ex14-binding is specific to DMC1 rings, we introduced an F85E mutation that targets the same residue of DMC1's self-association interface. Firstly, we confirmed that DMC1 ΔN F85E formed only monomers in solution (Supplementary Fig. 7). Then, through pulldowns we showed that whilst DMC1 ΔN F85E retained binding to BRC4, it failed to interact with either Ex14 or Ex14-Tr (Fig. 2e). Thus, we conclude that Ex14 selectively binds to oligomeric DMC1, via a PhePP-binding site that is formed by two divergent loops and depends on hydrophobic interactions of DMC1 residue V179.

## BRCA2 Ex14 stabilises DMC1 nucleoprotein filaments

What is the function of the BRCA2 Ex14-DMC1 interaction? We used electrophoretic mobility shift assays (EMSAs) to determine the impact of BRCA2-binding on DMC1 nucleoprotein filaments. We used two different conditions for nucleoprotein filament assembly. Firstly, a TAE (tris, acetate, EDTA; pH 7.5) buffer that was previously used for RAD51-ssDNA filaments[33], in which DMC1-ssDNA filaments were partially formed at the canonical ratio of three nucleotides to one protomer (Fig. 3a). Secondly, a TEA (triethanolamine; pH 7.5) + KCl buffer based on previous DMC1 studies[36–38], in which DMC1-ssDNA filament formation was more complete at the canonical ratio (Fig. 3b). Using the former unfavourable condition, we found that equimolar BRCA2 Ex14 stimulates DMC1-ssDNA nucleoprotein filament assembly,

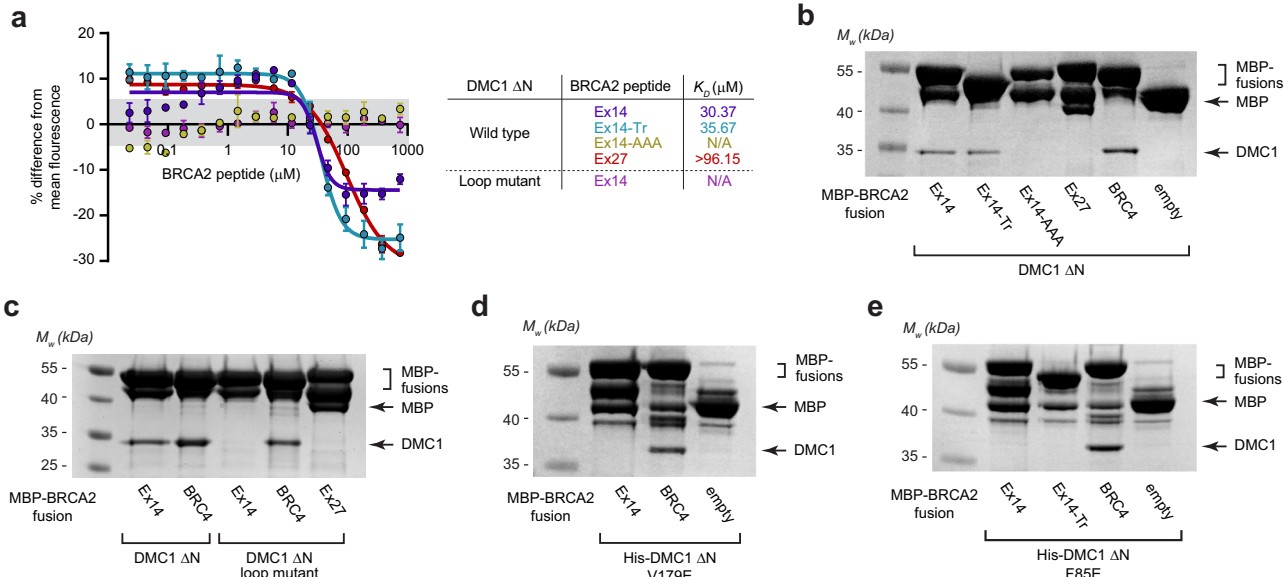

**Fig. 2 | BRCA2 Ex14 interacts with DMC1's oligomeric core via its PhePP-binding site. a** Microscale thermophoresis (MST) of interactions between BRCA2 peptides and DMC1 ΔN; data are presented as mean values, with error bars indicating standard error, *n* = 3 independent experiments. BRCA2 Ex14 and Ex14-Tr bind to DMC1 core with affinities of 30 μM and 36 μM, respectively, whereas interactions are not detectable for the BRCA2 Ex14-AAA mutant (F2406A, P2408A, P2409A), or for the DMC1 ΔN loop mutant (replacing 147-GAGGYPG-153 and 178-NVDHDA-183 with RAD51 amino-acids IDRGGGE and GLSGSD, respectively). BRCA2 Ex27 binds to DMC1 ΔN with an affinity of >96 μM. **b, c** Amylose pull-downs of (**b**) DMC1 ΔN, (**c**) DMC1 ΔN wild-type and loop mutant, following recombinant

co-expression with MBP-BRCA2 Ex14 wild-type, Ex14-Tr, Ex14-AAA mutant, Ex27, BRC4 and free MBP (empty). **d, e** Amylose pull-downs of (**d**) His-DMC1 ΔN V179E mutant and (**e**) His-DMC1 ΔN F85E mutant, following recombinant expression and mixing with recombinantly expressed MBP-BRCA2 Ex14 wild-type, Ex14-Tr, Ex27, BRC4 and free MBP (empty). The MBP-BRCA2 fusion proteins exhibited some degradation to free MBP and intermediate species (MBP fused to a partially degraded peptide), which was more pronounced for Ex14 and Ex27 than for Ex14-Tr and BRC4. **b–e** Gel images are representative of at least three replicates. Source data are provided as a Source Data file.

achieving complete DMC1-ssDNA filament formation at the canonical binding ratio (Fig. 3a). We observed the same stimulation of DMC1-ssDNA filament formation upon changing the order of addition of ssDNA, DMC1 and Ex14 (Supplementary Fig. 8). Using the latter favourable condition, we found that BRCA2 Ex14 binds to pre-formed DMC1-ssDNA filaments, inducing a super-shift (Fig. 3b). This was noticeable using Ex14 in equimolar amounts (11 μM), and became substantial at a 5-fold stoichiometric excess (55 μM) (Fig. 3b). As the $K_D$ of the DMC1-Ex14 interaction is approximately 30 μM (Fig. 2a), the requirement for a stoichiometric excess is likely due to binding being favoured at Ex14 concentrations above the $K_D$. The Ex14-AAA PhePP mutant largely failed to stimulate or super-shift DMC1-ssDNA filaments, albeit with some stimulation of filament formation at high concentrations (Fig. 3a, b), and filaments formed using DMC1 V179E underwent only minimal super-shift by Ex14 (Supplementary Fig. 9a–c), indicating that these functions depend on the interaction observed in the crystal structure. Further, truncated Ex14-Tr stimulated DMC1-ssDNA nucleoprotein filament formation but failed to super-shift pre-formed filaments (Fig. 3a, b).

Finally, we tested whether binding of BRCA2 Ex14 protects DMC1-ssDNA nucleoprotein filaments from BRC4-mediated disruption (Fig. 3c). We used the favourable conditions (TEA + KCl) for DMC1-ssDNA filament formation, and included Ex14 at a 10-fold stoichiometric excess (110 μM) to ensure that binding would be favoured by exceeding the $K_D$, and to match the conditions previously used for RAD51-Ex27[33]. Whilst DMC1-ssDNA filaments were completely disrupted by a 5-fold excess of BRC4, the presence of Ex14 protected filaments from disruption at a 10-fold excess of BRC4 (Fig. 3c). Notably, we also observed some protection from the same amount of BRC4 by equimolar Ex14, and complete protection by a 5-fold excess of Ex14 (Supplementary Fig. 10a), in keeping with the need to exceed its approximately 30 μM binding affinity. The protection of DMC1-ssDNA filaments from BRC4-mediated disruption was largely abrogated by

the Ex14-AAA PhePP mutation and was mostly lost for truncated Ex14-Tr (Fig. 3c). Hence, BRCA2 Ex14 binds to DMC1 via the PhePP motif, promoting the formation of DMC1-ssDNA nucleoprotein filaments, and protecting filaments from BRC4-mediated disruption, in the same way as was previously observed for the BRCA2 Ex27-RAD51 interaction[33,34]. Further, whilst the core PhePP interaction site is sufficient to stimulate DMC1-ssDNA nucleoprotein filament formation, surrounding amino-acids that are present within Ex14 (but not Ex14-Tr) are necessary to super-shift filaments and confer protection from BRC4-mediated disruption.

## BRCA2 Ex27 binds to RAD51 through a conserved interaction mode

We wondered whether BRCA2 PhePP motifs bind to RAD51 and DMC1 recombinases through the same interaction mode. Firstly, we built an *AlphaFold2* model of BRCA2 Ex27 bound to a RAD51 dimer (Supplementary Fig. 11a–e). We then built a model of BRCA2 Ex27 bound to a RAD51 filament by docking our BRCA2-RAD51 model onto a cryo-EM structure of the RAD51 filament (PDB accession 8BSC)[39]. We noticed that the N-terminal end of the modelled BRCA2 Ex27 peptide clashed with RAD51 self-association interactions, so removed this and restricted the extent of the Ex27 peptide to amino-acids 3278-3309 (Fig. 4a). The resultant model predicts that BRCA2 Ex27 binds via precisely the same core PhePP interface as observed in our BRCA2-DMC1 crystal structure (Fig. 4b, c). Further, the model predicts that the N-terminal end of BRCA2 Ex27 binds across RAD51's self-association interface, explaining how it protects filaments from BRC4-mediated disruption[33,34]. Hence, the N-terminal end of BRCA2 Ex14 may bind across the DMC1 self-association interface in a similar manner to protect from BRC repeat disruption.

We next validated our BRCA2-RAD51 model. Pull-down assays confirmed that RAD51 binds selectively to BRCA2 Ex27 rather than Ex14, in a manner that is disrupted by mutation of the PhePP motif

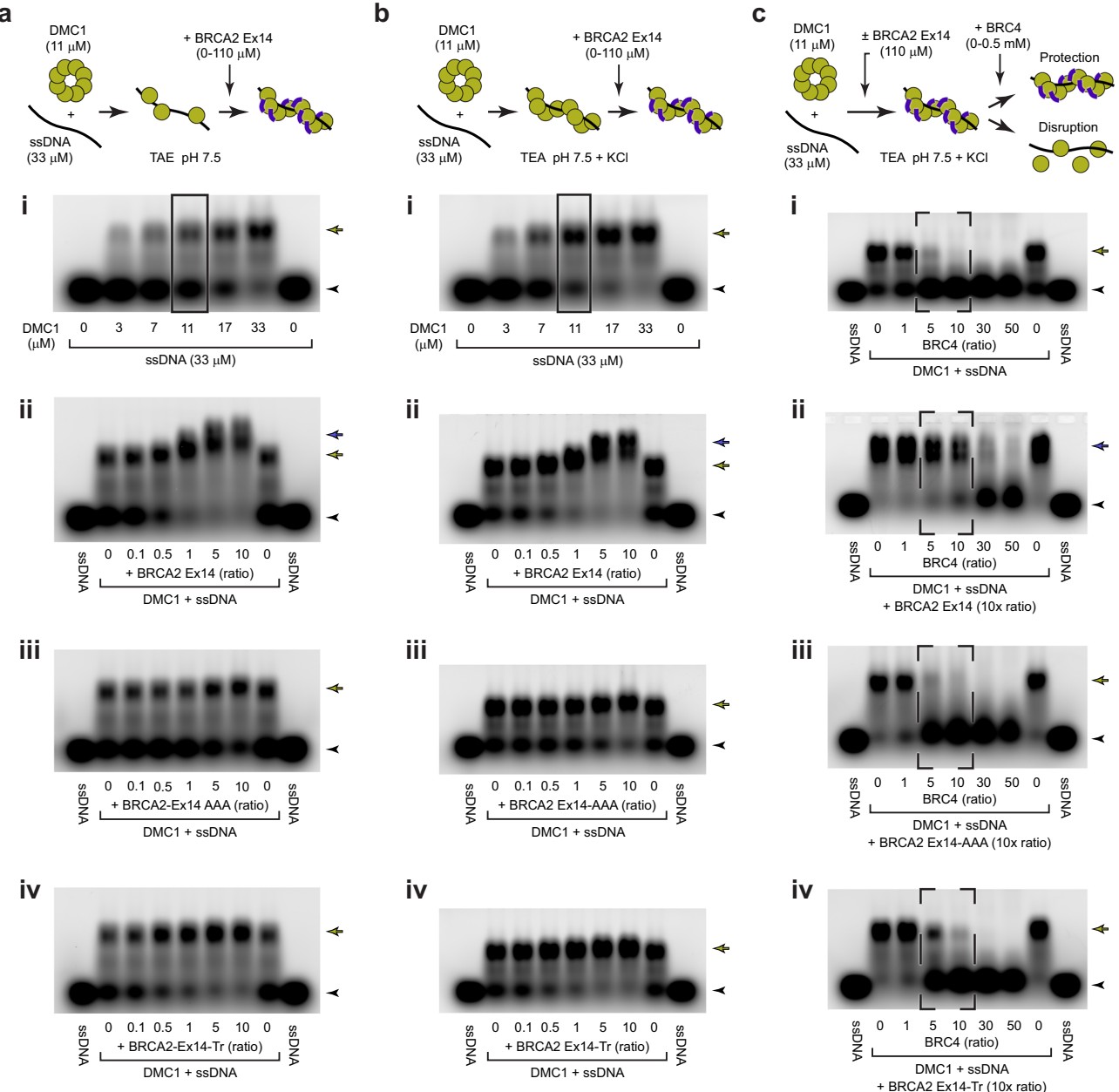

**Fig. 3 | BRCA2 Ex14 stabilises DMC1-ssDNA complexes.** Electrophoretic mobility shift assays (EMSAs) analysing the ability of BRCA2 Ex14 to bind, promote and protect DMC1-ssDNA filaments. **a** EMSAs using TAE pH 7.5 conditions in which (**i**) DMC1-ssDNA binding was incomplete at the canonical ratio of three nucleotides to one protomer (boxed). The canonical ratio (boxed) from subpanel (**i**) was used in subsequent EMSAs. (**ii**) BRCA2 Ex14 promoted the formation of DMC1-ssDNA filaments. (**iii**) The BRCA2 Ex14-AAA mutant largely failed to stimulate filament formation and (**iv**) truncated BRCA2 Ex14-Tr mostly stimulated DMC1-ssDNA filament formation. **b** EMSAs using TEA pH 7.5 + KCl conditions in which (**i**) DMC1-ssDNA binding was largely complete at the canonical ratio of three nucleotides to one protomer (boxed). The canonical ratio (boxed) from subpanel (**i**) was used in subsequent EMSAs. (**ii**) BRCA2 Ex14 induced a super-shift, demonstrating binding to DMC1-ssDNA filaments. The super-shift was largely eliminated by (**iii**) the BRCA2

Ex14-AAA mutant and (**iv**) truncated BRCA2 Ex14-Tr. The order of addition did not affect the ability of Ex14 to induce the formation and super-shift of DMC1-ssDNA filaments, as shown in Supplementary Fig. 8. **c** EMSAs using TEA pH 7.5 + KCl conditions in which (**i**) DMC1-ssDNA binding was disrupted by a stoichiometric excess of BRC4 (dashed, boxed). (**ii**) BRCA2 Ex14 protected against BRC4-mediated disruption (dashed, boxed). The protection was completely abrogated by (**iii**) the BRCA2 Ex14-AAA mutant, and was diminished in (**iv**) truncated BRCA2 Ex14-Tr. BRCA2 peptide concentrations are shown as molar ratios with respect to DMC1 protomers. Arrowheads, free ssDNA; yellow arrows, DMC1-ssDNA complexes; blue arrows, BRCA2-DMC1-ssDNA complexes. The ssDNA substrate is a 100-nucleotide random sequence (provided in Methods). **a**–**c** Gel images are representative of at least three replicates. Source data are provided as a Source Data file.

(herein referred to as BRCA2 Ex27-AAA; amino-acids F3308A, P3310A, P3311A) (Fig. 4d). Further, we found that protection of RAD51-ssDNA nucleoprotein filaments from BRC4-mediated disruption is abrogated by the BRCA2 Ex27-AAA mutant and by an L180E mutation that targets the divergent loop of RAD51's PhePP-binding site and is equivalent to the DMC1 V179E mutation (Fig. 4e). These assays were performed using

the same random sequence ssDNA substrate that was used for DMC1. However, it was previously found that protection of RAD51-ssDNA filaments by Ex27 is more complete for polydT ssDNA substrates[33]. Hence, we repeated these EMSAs using polydT ssDNA in which Ex27 conferred complete protection from disruption by BRC4 (Supplementary Fig. 12). Protection of RAD51-ssDNA filaments using polydT

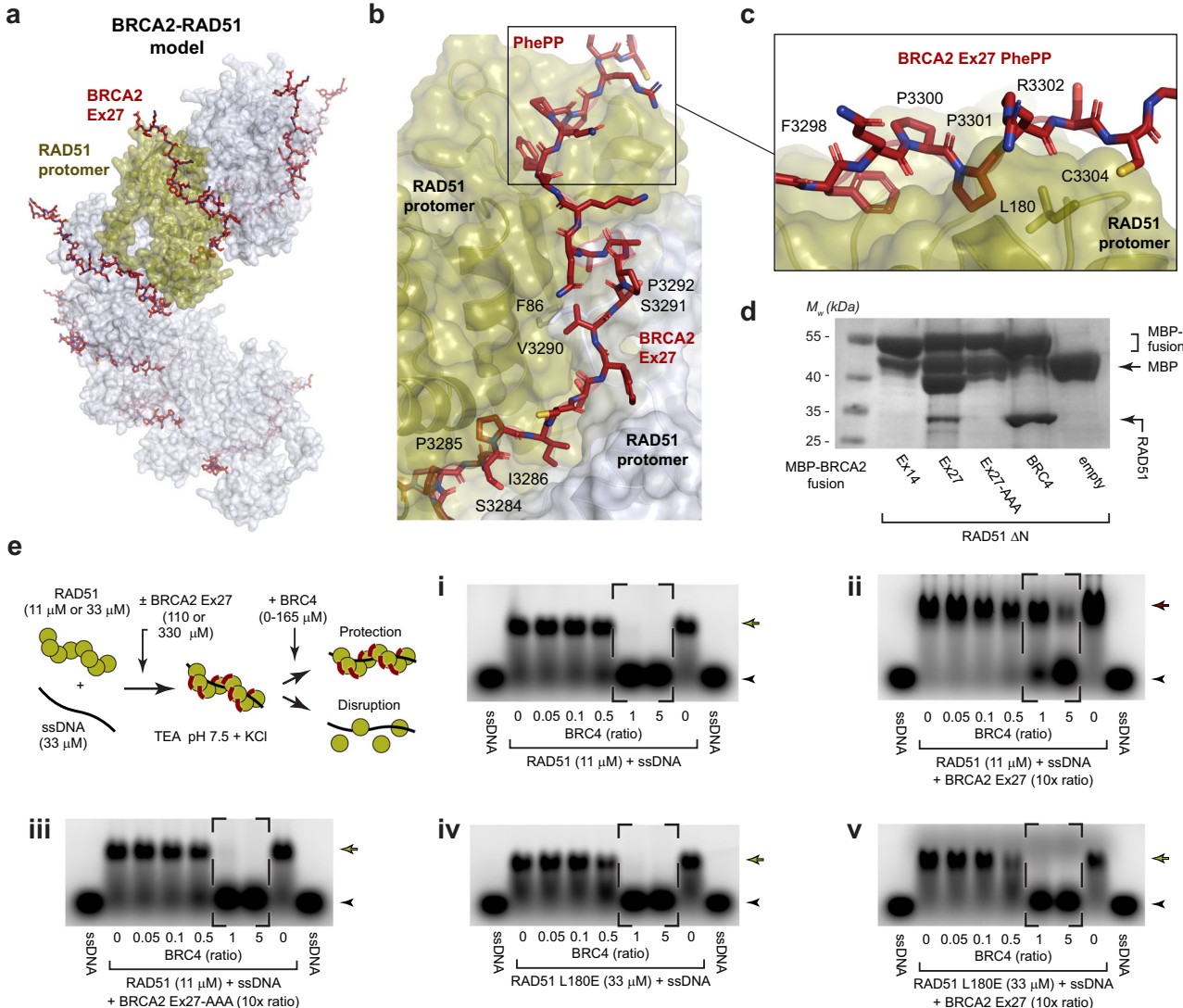

**Fig. 4 | BRCA2 Ex27 stabilises RAD51-ssDNA complexes. a** Model of the BRCA2 Ex27-RAD51 filament structure generated by docking BRCA2 Ex27-RAD51 1:2 complex *AlphaFold2* models (Supplementary Fig. 11) onto a previously reported structure of the RAD51 filament (PDB accession: 8BSC)[39]. **b** The modelled BRCA2 Ex27 peptide includes the PhePP motif (boxed), and a long extension that runs along the interface between adjacent RAD51 promoters, shrouding the F86 self-association interaction. **c** The PhePP motif is predicted to bind in the same manner as the BRCA2 Ex14-DMC1 interaction, involving FxPP amino-acids F3298, P3300 and P3301. **d** Amylose pull-downs of RAD51 ΔN following recombinant co-expression with MBP-BRCA2 Ex14, Ex27 and its AAA mutant (F3298A, P3300A, P3301A), BRC4 and free MBP (empty). The MBP-BRCA2 fusion proteins exhibited some degradation to free MBP and intermediate species (MBP fused to a partially degraded peptide), which was more pronounced for Ex14 and Ex27 than for BRC4. **e** EMSAs in which (**i**) RAD51-ssDNA binding was disrupted by equimolar quantities of BRC4 (dashed, boxed). (**ii**) BRCA2 Ex27 but not (**iii**) BRCA2 Ex14-AAA mutant protected against BRC4-mediated disruption (dashed, boxed), and (**iv**) RAD51 L180E mutant underwent similar disruption, but (**v**) was not protected by BRCA2 Ex27. BRCA2 peptide concentrations are shown as molar ratios with respect to RAD51 protomers. Arrowheads, free ssDNA; yellow arrows, RAD51-ssDNA complexes; red arrows, BRCA2-RAD51-ssDNA complexes. The ssDNA substrate is a 100-nucleotide random sequence (provided in Methods). The same EMSAs performed using a 100-nucleotide polydT ssDNA substrate, in which complete protection is conferred by Ex27 and abrogated upon mutation, are shown in Supplementary Fig. 12. **d, e** Gel images are representative of at least three replicates. Source data are provided as a Source Data file.

substrates was abrogated by the Ex27-AAA PhePP and RAD51 L180E mutations (Supplementary Fig. 12). Further, whilst these assays were performed using the same 10-fold excess of Ex27 as in previous studies[33], protection from a 10-fold excess of BRC4 was largely conferred by a 3-fold excess of Ex27 (33 μM), in keeping with RAD51-Ex27 having a similar binding affinity to DMC1-Ex14 (Supplementary Fig. 10b). Hence, BRCA2 Ex27 protects RAD51-ssDNA nucleoprotein filaments by binding to RAD51's PhePP-binding site in the same manner as observed in our BRCA2-DMC1 crystal structure. Thus, we conclude that BRCA2's PhePP motifs bind selectively to RAD51 and DMC1 nucleoprotein filaments through a conserved interaction mode that stabilises filaments from BRC4-mediated disruption.

## Discussion

The interaction mode between BRCA2's PhePP motifs and recombinases had remained unknown. Here, we have presented a crystal structure of the BRCA2-DMC1 complex, and solution data, in which we reveal a novel interaction mode, which is conserved between BRCA2's Ex14 and Ex27 PhePP motifs and recombinases DMC1 and RAD51, respectively. Further, similar PhePP motifs have been reported in other recombinases-interacting proteins, including RAD51AP1 and FIGNL1[40,41]. Hence, this may be a general mechanism whereby regulatory factors associate with RAD51 and DMC1 nucleoprotein filaments. Overall, our data support a dichotomy, in which BRCA2's BRC and PhePP motifs have distinct roles in remodelling recombinases and

stabilising nucleoprotein filaments, respectively, in both mitotic and meiotic recombination.

This study was performed using peptides corresponding to BRCA2's PhePP sequences. This approach was appropriate as the PhePP Ex14 and Ex27 sequences are located within unstructured regions of BRCA2, so likely operate as discrete functional modules. Hence, it is unlikely that other regions of BRCA2 substantially alter their functions. Whilst we lack experimental data regarding the roles of Ex14 and Ex27 within full-length BRCA2 in vitro, regions that include Ex14 and Ex27 have been targeted genetically in mice. Deletion of exon 27 led to shortened lifespan, hypersensitivity to DNA-damaging agents and chromosomal instability[42,43], with defective response to stalled or collapsed replication forks. A point mutation of mouse BRCA2 equivalent to F2406D remained fertile, suggesting that Ex14 may be dispensable[44]. However, deletion of exons 12-14 led to infertility with failure of DMC1 foci formation[45]. Whilst this deletion included both MEILB2-binding sites (exons 12-13) and Ex14, a comparable deletion of only exon 12 was fertile[46], suggesting that infertility may have resulted from loss of Ex14. Hence, Ex27 is necessary for BRCA2 function in vivo, whereas evidence regarding Ex14 remains uncertain. The following discussion assumes that Ex14 and Ex27 form the same interactions with DMC1 and RAD51, and confer the same functions regarding their nucleoprotein filaments, within the context of full-length BRCA2 in vivo.

How does BRCA2 PhePP-binding confer stability to RAD51 and DMC1 nucleoprotein filaments? The BRCA2-DMC1 crystal structure reported herein shows only the core PhePP interface, which is distant from the recombinase self-association interface. Further, whilst truncated Ex14 was sufficient to bind DMC1 and stimulate nucleoprotein filament formation, the wider sequence present within Ex14 was necessary to confer protection against BRC-mediated disruption. Hence, protection appears to depend on interactions beyond those observed in the BRCA2-DMC1 crystal structure. The BRCA2 Ex27-RAD51 model suggests that sequence upstream of the core PhePP interaction site binds across the self-association interface, which may sterically hinder access, thereby explaining its stabilisation and protection from BRC-mediated disruption[33,34]. In a parallel study, a cryo-EM structure of the BRCA2 Ex27-RAD51 complex was reported[47], which agrees with the model presented herein. Further, in the cryo-EM structure and in our model, the positively charged N-terminal end of Ex27 is located close to the filament axis, where bound DNA is located. Hence, we suggest that Ex27 further stabilises RAD51 nucleoprotein filaments by its N-terminus binding directly to DNA, in keeping with recent biochemical findings[48]. It is likely that BRCA2 Ex14 confers stability and protection to DMC1 nucleoprotein filaments in a similar manner to Ex27, through its upstream sequence binding across the self-association interface. Hence, the residual stimulation of DMC1-ssDNA filaments by Ex14-AAA at high concentrations (Fig. 3a, b) may be due to Ex14's upstream sequence binding directly across the self-association interface without being anchored by the PhePP core interaction site. However, it will require future structural analysis of the full Ex14-DMC1-ssDNA complex to determine whether and how upstream Ex14 amino-acids bind across DMC1's self-association interface.

The extensive BRCA2 Ex27-RAD51 interaction, which spans RAD51's self-association interface, provides clear rationale for how Ex27 specifically binds to oligomeric RAD51. However, it is difficult to explain how the shorter Ex14 sequence, and particularly truncated Ex14-Tr, specifically recognises oligomeric rather than monomeric DMC1. Indeed, only the core PhePP interaction site was observed in the crystal structure, and the Ex14-Tr contains only a few additional amino acids on either side of the observed interface. Hence, we suggest that specificity may be achieved through an allosteric change in DMC1's PhePP-binding site between its oligomeric and monomeric forms that enables binding only in the former case. In this regard, Ex14-binding may also allosterically favour the formation of ssDNA-bound filaments rather than unbound

rings, providing a possible explanation for how truncated Ex14-Tr stimulates DMC1-ssDNA filament formation despite the PhePP-binding site being distant from the self-association interface.

There are some notable differences in the sequences of BRCA2 Ex14 and Ex27, which must affect their function. Firstly, Ex14 shows little conservation and no residues that are critical for binding outside of the PhePP motif[32], suggesting that whilst it must extend across the self-association interface to confer protection, it forms fewer recombinase interactions than Ex27. Secondly, Ex14 lacks the CDK phosphorylation site that is present in Ex27 and is responsible for disrupting binding at the G2-M transition[27]. This likely represents the difference between the BRCA2 Ex14-DMC1 interaction functioning in meiosis, and the BRCA2 Ex27-RAD51 interaction functioning in replication fork preservation[4,5,48]. Thirdly, the core PhePP interaction site of Ex14 (2406-FVPPFK-2411) is more hydrophobic than the equivalent site of Ex27 (3298-FQPPRS-3303). The V2406 and Q3299 residues are entirely solvent-exposed, so it is not possible to rationalise how they may favour selective DMC1/RAD51-binding. In contrary, F2410 and R3302 have different orientations and interactions between Ex14-DMC1 and Ex27-RAD51 structures. F2410 (Ex14) packs against a hydrophobic surface of DMC1 that is formed by divergent loop A, which may not be possible in RAD51 owing to a salt bridge between R150 of loop A and a backbone carbonyl of loop B (Fig. 1d). R3302 (Ex27) forms a salt bridge with RAD51 residue D187[47], which would not be possible in DMC1 as its equivalent residue is an alanine. Hence, PhePP residues F2410 and R3302 may contribute towards the DMC1/RAD51-binding specificity of Ex14 and Ex27. Further, F2410 may enhance the binding affinity such that Ex14 requires only the core interaction site rather than the more extensive interface formed by Ex27.

Whilst BRCA2 PhePP interactions have thus far been considered in isolation, they likely operate as part of a wider 'recombinosome' structure. BRCA2's PhePP motifs are located on either side of its C-terminal DNA-binding domain[49]. Hence, filament-stabilising interactions by Ex14 and Ex27 may occur cooperatively with DNA-binding and nucleation of nucleoprotein filament formation by BRCA2, which is known to involve displacement of ssDNA-binding protein RPA[15,16]. Further, Ex14 is located adjacent to a binding site of meiotic recombination complex MEILB2-BRME1[50,51], which is proposed to displace meiosis-specific ssDNA-binding complex MEIOB-SPATA22 and act as DNA clamp[52,53]. Hence, BRCA2 may coordinate the formation of a 'meiotic recombinosome' in which the key functionalities of meiotic recombination are brought together to enable the loading and stabilisation of DMC1 nucleoprotein filaments for inter-homologue recombination. Our understanding of meiotic recombination would be truly transformed by structure elucidation of this multi-protein assembly.

## Methods

### Recombinant protein expression and purification
The sequences corresponding to human DMC1 (amino acids 1-340) and DMC1 ΔN (83-340) were cloned into pHAT4[54] for expression as a TEV-cleavable His6-tag fusion in BL21 (DE3) cells (Novagen®) in 2xYT media, induced with 0.5 mM IPTG for 16 hours at 25 °C. Cells were lysed by sonication in 20 mM Tris pH 8.0, 500 mM KCl and fusion proteins were purified from clarified lysate through consecutive Ni-NTA (Qiagen) and HiTrap Q HP (GE Healthcare) ion exchange chromatography. His6-tag removal was mediated by incubation overnight with TEV protease. Cleaved DMC1 was further purified by HiTrapQ HP ion exchange chromatography and size exclusion chromatography (HiLoad™ 16/600 Superdex 200, GE Healthcare) in 20 mM Tris pH 8.0, 150 mM KCl, 2 mM DTT.

Monomeric DMC1 ΔN (amino acids 83-340) harbouring the F85E mutation was cloned into pHAT4[54] for expression as a TEV-cleavable His6-tag fusion in BL21 (DE3) cells (Novagen®) in 2xYT media, induced with 0.5 mM IPTG for 16 hours at 25 °C. Cells were lysed by sonication

in 20 mM Tris pH 8.0, 500 mM KCl, 10% glycerol in the presence of benzonase. The His6-tag fusion protein was purified from clarified lysate through consecutive Ni-NTA (Qiagen) and size exclusion chromatography (HiLoad™ 16/600 Superdex 200, GE Healthcare) in 20 mM Tris pH 8.0, 500 mM KCl, 10 % glycerol, 2 mM DTT.

RAD51 (amino acids 1–339) and RAD51 L180E were co-expressed with an MBP-fusion of BRCA2 BRC4 (amino acids 1519-1551) in BL21 (DE3) cells (Novagen®) in 2xYT media, and induced with 0.5 mM IPTG for 16 hours at 25 °C. Cells were lysed by sonication in 20 mM Tris pH 8.0, 500 mM KCl, and the complex of RAD51 and MBP-BRCA2 BRC4 was purified from clarified lysate through consecutive amylose and Ni-NTA (Qiagen) affinity chromatography. The complex was disrupted through HiTrap Q HP (GE Healthcare) ion exchange chromatography, and RAD51 was further purified through HiTrap Heparin HP (GE Healthcare) ion-exchange chromatography and size exclusion chromatography (HiLoad™ 16/600 Superdex 200, GE Healthcare) in 20 mM Tris pH 8.0, 150 mM KCl, 2 mM DTT.

The sequences corresponding to human BRCA2 Ex14 (amino acids 2387-2420) and BRCA2 Ex14-AAA were cloned into pMAT11 for expression as TEV-cleavable His6-MBP-tag fusions in BL21 (DE3) cells (Novagen®) in 2xYT media, induced with 0.5 mM IPTG for 16 hours at 25 °C. Cells were lysed by sonication in 20 mM Tris pH 8.0, 500 mM KCl and the fusion proteins purified from clarified lysate through consecutive Ni-NTA (Qiagen) and HiTrap Q HP (GE Healthcare) ion exchange chromatography. His6-MBP-tag removal was mediated by incubation overnight with TEV protease. Cleaved BRCA2 Ex14 was further purified by HiTrapSP HP ion exchange chromatography and size exclusion chromatography (HiLoad™ 16/600 Superdex 200, GE Healthcare) in 20 mM Tris pH 8.0, 150 mM KCl, 2 mM DTT.

All other BRCA2 peptide sequences utilised in experiments (other than affinity pulldown experiments) were synthesised by Severn Biotech including BRC4 (amino acids 1519-1551), BRCA2 Ex14-Tr (amino acids 2401-2414), BRCA2 Ex27 (amino acids 3270-3315) and BRCA2 Ex27-AAA (amino acids 3270-3315 harbouring F3308A, P3310A and P3311A mutations).

Protein samples were concentrated using 10,000/3000 MWCO centrifugal units (Amicon) and were stored at −80 ˚C following flash-freezing in liquid nitrogen. Protein samples were analysed by SDS-PAGE with Coomassie staining, and concentrations were determined by UV spectroscopy using a Cary 60 UV spectrophotometer (Agilent) with extinction coefficients and molecular weights calculated by ProtParam (http://web.expasy.org/protparam/).

For affinity pulldown experiments, MBP-fusions of BRCA2 fragments (Ex14, amino-acids 2387-2420, Ex14-Tr, amino acids 2401-2414, Ex27, amino acids 3270-3315, BRC4, amino acids 1519-1551) were cloned into pMAT11 vectors[54] and co-expressed with DMC1 ΔN, DMC1 ΔN loop mutant (in which DMC1 amino acids 147-GAGGYPG-153 and 178-NVDHDA-183 were replaced by the cognate sequences of RAD51 148-IDRGGGE-154 and 179-GLSGSD-184), RAD51 ΔN, RAD51 ΔN loop mutant (the reverse switch of DMC1 ΔN loop mutant), and RAD51. MBP-fusions and complexes were purified through amylose affinity chromatography.

For affinity experiments involving DMC1 ΔN V179E and F85E, MBP-fusions of BRCA2 fragments (Ex14, amino-acids 2387-2420, Ex14-Tr, amino acids 2401-2414, BRC4, amino acids 1519-1551) were cloned into pMAT11 vectors and DMC1 ΔN mutants (V179E or F85E) were cloned into pHAT4 vectors[54]. Lysates of bacterial cultures expressing His6-DMC1 mutants or MBP-BRCA2 constructs were mixed, and complexes were purified through amylose affinity chromatography in 20 mM Tris pH 8.0, 500 mM KCl, 2 mM DTT and eluted using 30 mM D-maltose.

### Microscale thermophoresis

DMC1 ΔN and DMC1 ΔN loop mutant were labelled in 10 mM HEPES pH 8.0, 150 mM NaCl with an amine reactive fluorescent dye (Monolith™ Protein Labelling Kit BLUE-NHS, Nanotemper GmbH). A serial dilution of titrant (peptide) was prepared in 10 mM HEPES pH 8.0,

150 mM NaCl, 0.05 % Tween-20. The concentration of labelled DMC1 ΔN or DMC1 ΔN loop mutant were constant at 150 nM and the concentration of the interacting peptide was titrated in 1:1 dilutions starting from 750 μM. An equal volume (10 μl) of the serial dilution and of the diluted labelled DMC1 were mixed and loaded in glass capillaries (Monolith™ NT.115 MST Premium Coated Capillaries, Nanotemper GmbH). The initial fluorescence and the thermophoretic mobility were measured on a Monolith NT.115 system (NanoTemper Technologies). The excitation power was 45 % and MST power was 60 % at room temperature (-22 °C). Laser on and off times were set at 5 and 30 seconds, respectively. Data analysis was performed with NT Affinity Analysis software.

As a change in initial fluorescence was detected, SDS denaturation (SD) tests were performed according to the manufacturer's instructions to determine whether these changes were ligand-induced and whether the data could therefore be used to derive affinity measurements. Briefly, 7 μl of the samples containing the three highest and three lowest concentrations of peptide were mixed with 7 μl of denaturation buffer (4 % SDS, 40 mM DTT). After incubating each for 5 minutes at 95 °C to denature the proteins and subject to brief centrifugation, the samples were loaded onto Premium Coated Capillaries and their initial fluorescence measurements recorded. Here, an equalisation of initial fluorescence readings during the SD test demonstrate that the change in fluorescence is ligand-dependent and the initial fluorescence data can be used to derive a binding affinity. Initial fluorescence readings were used to derive binding affinities.

### Crystallisation and structure solution of DMC1

DMC1 ΔN protein crystals were obtained through vapour diffusion in hanging drops, by mixing 1 μl of protein at 15 mg/ml with 1 μl of crystallisation solution (50 mM HEPES-NaOH pH 7.2, 50 mM MgCl₂, 500 mM NaCl, 7.5 % PEG3350) and equilibrating at 20 °C for 2 weeks. Crystals were cryo-protected using 6 M sodium formate and cryo-cooled in liquid nitrogen. X-ray diffraction data were collected at 0.9159 Å, 100 K, as 2000 consecutive 0.10° frames of 0.010 s exposure on a Pilatus 6 M detector at beamline I04-1 of the Diamond Light Source synchrotron facility (Oxfordshire, UK). Data were indexed, integrated in *XDS*[55], scaled in *XSCALE*[56], and merged in *Aimless*[57]. Crystals belong to orthorhombic spacegroup I422 (cell dimensions $a = 174.49$ Å, $b = 174.49$ Å, $c = 179.57$ Å, $\alpha = 90°$, $\beta = 90°$, $\gamma = 90°$), with four DMC1 protomers in the asymmetric unit. Structure solution was achieved through molecular replacement using *PHASER*[58], with a single chain from PDB accession 4HYY as a search model. Model building was performed through iterative re-building by *PHENIX Autobuild*[59] and manual building in *Coot*[60], with the addition of PEG ligands. The structure was refined using *PHENIX refine*[59], using isotropic atomic displacement parameters with two TLS groups per chain. The structure was refined against data to 2.05 Å resolution, to $R$ and $R_{free}$ values of 0.1942 and 0.2191 respectively, with 99.07% of residues within favoured regions of the Ramachandran plot (0 outliers), clashscore of 1.28 and overall *MolProbity* score of 0.85[61].

### Crystallisation and structure solution of a BRCA2-DMC1 complex

DMC1 ΔN protein crystals were obtained through vapour diffusion in hanging drops, by mixing 1 μl of protein at 20 mg/ml with 1 μl of crystallisation solution (50 mM HEPES-NaOH pH 7.4, 50 mM MgCl₂, 500 mM NaCl, 8 % PEG 3350) and equilibrating at 20 °C for 2 weeks. Crystals were soaked overnight in crystallisation solution containing 3.5 mM BRCA2 Ex14-Tr peptide (2401-RPTKVFVPPFKTKS-2414; synthesised by Severn Biotech). Crystals were cryo-protected using 20 % glycerol and cryo-cooled in liquid nitrogen. X-ray diffraction data were collected at 0.9795 Å, 100 K, as 2000 consecutive 0.10° frames of 0.050 s exposure on a Pilatus3 6 M detector at beamline I04 of the Diamond Light Source synchrotron facility (Oxfordshire, UK). Data

were processed using *AutoPROC*[62], in which indexing, integration, scaling and merging were performed by *XDS*[55] and *Aimless*[57], and anisotropic correction with a local I/σ(I) cut-off of 1.2 was performed by *STARANISO*[63]. Crystals belong to tetragonal spacegroup P4$_1$22 (cell dimensions $a = 125.40$ Å, $b = 125.40$ Å, $c = 364.195$ Å, $\alpha = 90°$, $\beta = 90°$, $\gamma = 90°$), with one DMC1 octamer in the asymmetric unit. Structure solution was achieved through molecular replacement using *Phaser*[58], through placement of four DMC1 dimers from its high-resolution structure (PDB accession 6R3P). The intact molecular replacement solution was refined using *PHENIX refine*[59] and alternative side-chain conformations were removed. BRCA2 peptides were built using *AlphaFold2* models of a BRCA2-DMC1 1:2 complex, which were docked onto the DMC1 protomers of the octameric ring using *PyMOL*. The completed BRCA2-DMC1 structure was refined using *PHENIX refine*[59], with isotropic atomic displacement parameters, using reference model restraints from the high-resolution DMC1 structure (PDB accession 6R3P). The structure was refined against anisotropy-corrected data with resolution limits between 3.4 Å and 5.8 Å, to $R$ and $R_{free}$ values of 0.2635 and 0.3057 respectively, with 98.80% of residues within the favoured regions of the Ramachandran plot (0 outliers), clashscore of 2.88 and overall *MolProbity* score of 1.08[61].

### Electrophoretic mobility shift assays (EMSAs)

To form nucleoprotein filaments, DMC1 and RAD51 was incubated with fluorescent 100-base ssDNA (5'-FAM- AATTCTCATTTTACTTACCGGA CGCTATTAGCAGTGGGTGAGCAAAAACAGGAAGGCAAAATGCCGCAAA AAAGGGAATAAGGGCGACACGGAAATGTTG-3') at the concentrations indicated in the figures for 30 minutes at 4 °C in reaction buffer. For DMC1, reaction buffer is 25 mM TEA (triethanolamine) pH 7.5, 1 mM ATP, 2 mM MgCl2, 0.1 mg/ml BSA, 2 mM DTT, 200 mM KCl. For RAD51, reaction buffer is 50 mM TEA (triethanolamine), 2 mM ATP, 0.5 mM MgCl$_2$, 2 mM DTT. For assays requiring incomplete nucleoprotein filament assembly, the reaction buffer consisted of 50 mM TAE (tris, acetate, EDTA) pH 7.5, 2 mM ATP, 2 mM MgCl2, 2 mM DTT. In nucleoprotein formation stimulation assays or supershift experiments, BRCA2Ex14 or BRCA2Ex14-AAA were added for a further 30 min at 4 °C. For protection experiments, the DMC1, RAD51, or RAD51 L180E–DNA complexes were incubated with BRCA2 Ex14, Ex14-AAA, Ex27, or Ex27-AAA for 30 min at 4 °C prior to the addition of BRC4 and further incubation for 30 min at 4 °C (concentrations provided in figures). Glycerol was added at a final concentration of 3%, and samples were analysed by electrophoresis on a 1% (w/v) (for DMC1) or 0.66% (w/v) (for RAD51) agarose gel in 0.5x TAE pH 8.0 at 20–40 V for ~4 h at 4 °C. DNA was detected by FAM using a Typhoon™ FLA 9500 (GE Healthcare), with 473 nm laser at excitation wavelength 490 nm and emission wavelength 520 nm, using the LPB filter and a PMT voltage of 500 V. To analyse RAD51 nucleoprotein filaments assembled upon 100-base poly(dT) ssDNA, gels were stained post-run using SYBR™ Gold Nucleic Acid Gel Stain (Invitrogen) and detected using a Typhoon™ FLA 9500 (GE Healthcare).

### Size-exclusion chromatography multi-angle light scattering (SEC-MALS)

The absolute molecular masses of wild-type and mutated DMC1 proteins were determined by size-exclusion chromatography multi-angle light scattering (SEC-MALS). Wild-type DMC1 and DMC1 harbouring the "loop" and V179E mutations were loaded onto a Superdex™ 200 Increase 10/300 GL size exclusion chromatography column (GE Healthcare) in 20 mM Tris pH 8.0, 150 mM KCl, 2 mM DTT, at 0.5 ml/min using an ÄKTA™ Pure (GE Healthcare). Monomeric DMC1 ΔN (amino acids 83-340) was analysed in the same manner using buffer containing 20 mM Tris pH 8.0, 500 mM KCl, 10 % glycerol, 2 mM DTT. The column outlet was fed into a DAWN® HELEOS™ II MALS detector (Wyatt Technology), followed by an Optilab® T-rEX™ differential refractometer (Wyatt Technology). Light scattering and differential refractive index data were collected and analysed using ASTRA® 6 software (Wyatt Technology). Molecular weights and estimated errors were calculated across eluted peaks by extrapolation from Zimm plots using a dn/dc value of 0.1850 ml/g. SEC-MALS data are presented as differential refractive index (dRI) profiles, with fitted molecular weights ($M_W$) plotted across elution peaks.

### Size-exclusion chromatography small-angle X-ray scattering

Size-exclusion chromatography small-angle X-ray scattering experiments were performed at beamline B21 of the Diamond Light Source synchrotron facility (Oxfordshire, UK). Protein samples at concentrations >5 mg/ml were loaded onto a Superdex™ 200 Increase 10/300 GL size exclusion chromatography column (GE Healthcare) in 20 mM Tris pH 8.0, 500 mM KCl at 0.5 ml/min using an Agilent 1200 HPLC system. The column outlet was fed into the experimental cell, and small-angle X-ray scattering data were recorded at 12.4 keV, detector distance 4.014 m, in 3.0 s frames. Data were subtracted and averaged, and analysed for Guinier region *Rg* and cross-sectional *Rg* (*Rc*) using *ScÅtter* 3.0 (http://www.bioisis.net), and *P(r)* distributions were fitted using *PRIMUS*[64]. Ab initio modelling was performed using *GASBOR*[65,66] in which 12-15 independent runs were performed in P8 symmetry and averaged. Crystal structures and models, in which missing loops were added using the *ModLoop* module of *MODELLER*[67] were docked into molecular envelopes using *SUPCOMB*[68], and were fitted to experimental data using *CRYSOL*[69].

### Structural modelling

Models were generated using a local installation of *AlphaFold2*[70]. Models of BRCA2-DMC1 and BRCA2-RAD51 complexes were generated using a 1:2 ratio of BRCA2 Ex14/Ex27 and DMC1 ΔN and RAD51, respectively. BRCA2-RAD51 models were docked onto a RAD51 filament (PDB accession: 8BSC)[39], and clashing regions at the N-termini (up to amino-acid 3277) of modelled BRCA2 Ex27 peptides were removed. For comparison between PhePP-binding loops, the BRCA2-DMC1 crystal structure was superposed with the RAD51 ΔN (PDB accession 1N0W)[31] using *PyMOL*.

### Protein sequence and structure analysis

Multiple sequence alignments were generated using *Jalview*[71], and molecular structure images were generated using the *PyMOL* Molecular Graphics System, Version 2.0.4 Schrödinger, LLC.

### Reporting summary

Further information on research design is available in the Nature Portfolio Reporting Summary linked to this article.

## Data availability

Crystallographic structure factors and atomic co-ordinates have been deposited in the Protein Data Bank (PDB) under accession numbers 6R3P and 8R2G, and raw diffraction data have been uploaded to https://proteindiffraction.org/. This study used PDB entries 1N0W, 4HYY and 8BSC. All remaining data that support the findings of this study are provided in the manuscript file and its supplementary information files. Source data are provided with this paper.

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

## Acknowledgements

We thank Diamond Light Source and the staff of beamlines I04 and I04-1 (proposals mx13587 and mx18598). We thank A. Baslé and J. Nicol for assistance with X-ray crystallographic data collection and DMC1 F83E analysis, respectively. J.M.D. is a Herchel Smith Fellow. O.R.D. is a Wellcome Senior Research Fellow (Grant Number 219413/Z/19/Z) and was supported by the Structural Biology core of the Wellcome Discovery Research Platform for Hidden Cell Biology (226791)

## Author contributions

J.M.D. performed experimental work. O.R.D. solved the crystal structures and wrote the manuscript. J.M.D. and O.R.D. designed experiments and analysed data.

## Competing interests

The authors declare no competing interests.
