## [Peer Review File · Nature Communications]

BRCA2 stabilises RAD51 and DMC1 nucleoprotein filaments through a conserved interaction modeREVIEWER COMMENTS

Reviewer #1 (Remarks to the Author):

RAD51 and its meiotic specific paralogue DMC1 are two central players of the Homologous Recombination (HR) process in meiosis. They form nucleoprotein filaments, by interacting with the single strand DNA generated after the resection of the 5' end of the DNA double strand break and, catalyze the strand exchange reaction followed by the invasion of the homologous duplex. However, illegitimate recombination might lead to genetic instability, increased in cancer risk and infertility and these proteins has to be tightly regulated.

In this paper, Duncce and Davies address the fundamental question of the regulation of RAD51/DMC1 nucleofilament formation by the BRCA2 tumor suppressor protein. BRCA2 directly interacts with RAD51/DMC1 proteins through BRC and PhePP motifs, promoting the formation of inactive or active filaments, respectively.

The authors propose a structural characterization: crystal structure of low resolution implemented by AlphaFold II, of the PhePP Ex14 motif with DMC1, as well as an AlphaFold II analysis of the BRCA2 Ex27 / RAD51 interaction. They show by EMSA how the two PhePP motifs, Ex14 and Ex27, interact specifically with DMC1 and RAD51 respectively, preassembled on ssDNA. They propose, using EMSA again, an extensive characterization of Ex14 and Ex27 interaction with DMC1 and RAD51 on a preassembled nucleofilament, in competition with the BRC motif of BRCA2. They also show how PhePP motifs stimulate DMC1 and RAD51 binding and how they protect the formed filaments against BRC dismantling. Finally, they propose from their structural analysis a conserved mode of interaction of BRCA2 PhePP motifs with DMC1 and RAD51, which can be extended to other repair proteins.

In general, this paper is very interesting and it addresses a key question for the field of HR in meiosis, with implication for a larger audience, considering the importance of HR outside of meiosis. It is well written, easy to follow, and the experiments well conducted. Structural analyses are pivotal to better understand the molecular mechanism underlying HR regulation, and this study gives important insight to better understand BRCA2 mode of action and regulation.

One main concern regarding the overall study and the understanding of BRCA2 mode of action in vivo comes from the choice of the BRCA2 peptides used, which are very small. One might wonder how is regulated the protein activity when all the BRCA2 domains are present together, in a more physiological context. Did the authors purify larger peptides that include the different domains of the protein together and test it for structural characterization and EMSA for interaction with preassembled DNA? If the authors would provide such analysis, they could start to answer a complex biological question of how BRCA2 is regulated and how it chooses its different mode of action. In addition, can they mention in vivo studies of mutants in the tested motifs? However, I understand that purification and analysis of larger peptides might be very complex, and I consider this not essential for acceptance of the work, even if it would be very interesting to add such data or comments on the implication. Altogether, I think that this manuscript is suitable for publication in

Nature Communications, after minor revisions which are listed below.

Major points:

On the different gels of Figure 1f to h, as well as on Fig 3d, the MBP fusions appear as multiple bands. What are the different species? The authors should give information on that.

From Figure 1, I am confused with the choice regarding the loops mutant, why not an unrelated loops to show the specificity of the interaction with the DMCA loop?

Regarding the stabilization of the filament through Ex14 interaction and the use of the AAA mutant. To me the effect of the AAA mutant is not complete (Figure 2a & 2b for example), suggesting additional effect or missing information. The authors should comment on this or be more precocious with the interpretations.

In the different experiments, the implication of BRCA2-Ex14 and BRCA2-Ex27 is shown on preassembled filaments. First, the conditions choose for preassembled (DMC1 11 μ M/ssDNA 33 μ M Fig2a for example) do not appear similar to me on the different gels, band intensities do not look similar? Second, what about the effect of BRCA2 Ex14 on a non-preassembled filament? Is it still possible to detect a stimulation, for example by modifying the kinetic observed, and if no, what is the interpretation? I speculate that this might have implication for the in vivo activity interesting to comment.

Regarding protection of RAD51 filament by BRCA2 Ex27 against BRC4 dismantling, is it a total protection? On Figure 3e ii, the effect of BRC4 is still visible.

Moreover, I am puzzled by the ratio used for the experiment, how can this huge excess of BRCA2 Ex27 can be interpreted, in particular in a physiological context where the different domains are present on the same protein? Did the authors test different ratios?

Minor points:

Figure 1e to f: strictly, these experiments do not show that Ex27 interacts with RAD51, which is shown in Fig3, and the literature, the authors should comment on that, or add RAD51 / Ex27 interaction on the gel, to simplify for the reader.

In Figure 3d and e: the gels contrast look very strange to me, with complete blank portion. I am sincerely convinced that the authors did not want to hide any information, however, even of low quality, a gel closer to reality would have been more informative, in particular if residual bands are present, and this should be commented.

Last section of the result, spelling mistake: BRC2 Ex27 has to be change to BRCA2 Ex27.

Reviewer #2 (Remarks to the Author):

The paper by Dunce and Davies reports the crystal structure of the meiotic recombinase DMC1 bound to its specific BRCA2-interaction motif PhePP, together with accompanying biochemical validation of the structure, and modelling of the similar RAD51 - BRCA2 PhePP interaction. The authors find that DMC1 and RAD51 interact in the same way with their respective BRCA2 PhePP motifs, and propose a unified model for how the interaction stabilises the filaments formed by the recombinases on DNA.

This is a nice and timely paper that reports an important advance in our understanding of BRCA2-dependent regulation of meiotic recombination, in agreement with complementary work that has been published very recently (Appleby et al, Nat Comm, 2023) and deposited in bioRxiv (Miron et al, 2023).

I would like to ask the authors to comment on the point below and implement the following minor changes before publication:

The DMC1-binding PhePP motif (FVPPF) is considerably more hydrophobic than the RAD51-binding one, with conservation of a solvent-exposed valine and a second phenylalanine. The additional phenylalanine in particular seems engaged in contacts with DMC1, from Fig. 1c, although the authors do not comment on these. Could this difference help explain the binding preference of DMC1 and RAD51 for their respective PhePP motifs?

ABSTRACT

The abstract is slightly repetitive and could be improved. Rather than stating once at the start: “BRCA2’s BRC repeats bind and disrupt RAD51 and DMC1 filaments, whereas its PhePP motifs binds to recombinases in a manner that stabilises their nucleoprotein filaments.” and then at the end: “In both cases, BRCA2 PhePP motifs enhance the stability of nucleoprotein filaments, protecting them from BRC-mediated disruption.”, it would be more informative for the reader to replace the second sentence with one explaining how the authors' findings provide insight into the stabilisation of the the DMC1/RAD51 filament by the PhePP motif.

INTRO

In line 4 of the Introduction, please rephrase “of which the machinery” so that the meaning of the sentence is clearer.

In the introduction, the assertion that isolated RAD51 form large filamentous structure is not quite accurate, please change to ‘heterogeneous oligomers’.

RESULTS

Correct typo: “The PhePP-binding site of DMC1 is formed from by two loops”.

Correct typo and clarify meaning of 'differential PhePP-binding site' in "The second loop includes amino-acid residues V179 (DMC1) and L180 (RAD51) that substantially contribute to the hydrophobic pocket of the differential PhePP-binding site (Figure 1d)."

Briefly explain why different buffers represent favourable and unfavourable conditions of filament formations, in "BRCA2 Ex14 stabilises DMC1 nucleoprotein filaments".

METHODS

I assume that the code in brackets at the end of the subheading refers to the PDB accession code. This might confuse the reader, please remove and provide accession codes under 'data availability'.

Please give residue numbers for the BRCA2 Ex14 peptide used for crystal soaking.

Reviewer #3 (Remarks to the Author):

In this manuscript, Duncanson and Davies used crystallography, aided by AlphaFold modeling, to determine the structure of recombinase DMC1 bound to a BRCA2-Ex14 peptide. They show in vitro that binding of BRCA2-Ex14 stimulates DMC1 filament assembly and provides protection against destabilization by the BRCA2-BRC domain. Similarly, RAD51 binding to BRCA2-Ex27 protects against BRC-mediated disruption.

The manuscript provides interesting insights into the role of BRCA2 in mitotic and meiotic recombination. I have a few comments that should be addressed prior to publication.

One limitation of the manuscript is the functional validation of the reported interface. The DMC1 loop mutant involves 12 mutations in total. A more careful residue-by-residue mutational analysis of the interface would be more appropriate. In particular, the importance of DMC1 residue V179 on binding to Ex14 and protection against BRC-mediated disruption should be established.

The authors show that Ex14 does not bind a monomeric DMC1 mutant, and hypothesize that residues upstream the PhePP motif bind across the self-association interface, similarly to the recently reported RAD51-Ex27 interaction (Appleby 2023).

However, the crystal structure uses a truncated version of Ex14 that lacks those upstream sequences. The authors should test the prediction that DMC1-F85E binds the Ex14-Tr peptide, and that DMC1 binding to Ex17-Tr does not confer protection against BRC-mediated disruption. Finally, independent analysis of the interaction mode of DMC1-Ex14 (and RAD51-Ex27) across the self-association interface, for example through cysteine crosslinking, would be relevant.

Minor points:

Is the model in Fig SF1 identical to the AlphaFold prediction in SF2? If not, a comparison of the

predicted model and experimental structure would be helpful.

A reference to the DMC1-F85E mutant should be provided.

Typo at the top of page 8: BRC2 Ex27.

Response to reviewers' comments
'BRCA2 stabilises RAD51 and DMC1 nucleoprotein filaments through a conserved interaction mode'
(NCOMMS-23-56434)

We are pleased that the reviewers recognise the value of this work, and are grateful for their thoughtful and considered comments. Here, we provide responses to the questions and comments raised by the reviewers, and outline how, in accordance with their requests, we have revised the manuscript.

Reviewer #1 (Remarks to the Author):

RAD51 and its meiotic specific paralogue DMC1 are two central players of the Homologous Recombination (HR) process in meiosis. They form nucleoprotein filaments, by interacting with the single strand DNA generated after the resection of the 5' end of the DNA double strand break and, catalyze the strand exchange reaction followed by the invasion of the homologous duplex. However, illegitimate recombination might lead to genetic instability, increased in cancer risk and infertility and these proteins has to be tightly regulated.

In this paper, Duncie and Davies address the fundamental question of the regulation of RAD51/DMC1 nucleofilament formation by the BRCA2 tumor suppressor protein. BRCA2 directly interacts with RAD51/DMC1 proteins through BRC and PhePP motifs, promoting the formation of inactive or active filaments, respectively.

The authors propose a structural characterization: crystal structure of low resolution implemented by AlphaFold II, of the PhePP Ex14 motif with DMC1, as well as an AlphaFold II analysis of the BRCA2 Ex27 / RAD51 interaction. They show by EMSA how the two PhePP motifs, Ex14 and Ex27, interact specifically with DMC1 and RAD51 respectively, preassembled on ssDNA. They propose, using EMSA again, an extensive characterization of Ex14 and Ex27 interaction with DMC1 and RAD51 on a preassembled nucleofilament, in competition with the BRC motif of BRCA2. They also show how PhePP motifs stimulate DMC1 and RAD51 binding and how they protect the formed filaments against BRC dismantling. Finally, they propose from their structural analysis a conserved mode of interaction of BRCA2 PhePP motifs with DMC1 and RAD51, which can be extended to other repair proteins.

In general, this paper is very interesting and it addresses a key question for the field of HR in meiosis, with implication for a larger audience, considering the importance of HR outside of meiosis. It is well written, easy to follow, and the experiments well conducted. Structural analyses are pivotal to better understand the molecular mechanism underlying HR regulation, and this study gives important insight to better understand BRCA2 mode of action and regulation.

One main concern regarding the overall study and the understanding of BRCA2 mode of action

in vivo comes from the choice of the BRCA2 peptides used, which are very small. One might wonder how is regulated the protein activity when all the BRCA2 domains are present together, in a more physiological context. Did the authors purify larger peptides that include the different domains of the protein together and test it for structural characterization and EMSA for interaction with preassembled DNA? If the authors would provide such analysis, they could start to answer a complex biological question of how BRCA2 is regulated and how it chooses its different mode of action. In addition, can they mention *in vivo* studies of mutants in the tested motifs? However, I understand that purification and analysis of larger peptides might be very complex, and I consider this not essential for acceptance of the work, even if it would be very interesting to add such data or comments on the implication. Altogether, I think that this manuscript is suitable for publication in *Nature Communications*, after minor revisions which are listed below.

1. As the reviewer highlights, this study used peptides corresponding to BRCA2's PhePP sequences. This was appropriate as Ex14 and Ex27 are conserved peptide sequences that are located within otherwise unstructured regions of BRCA2, so likely operate as independent functional units rather than being part of wider structure. The reviewer raises an important question regarding how the activities of such peptide sequences are regulated when part of the wider molecule. Hence, it would be ideal to study the structure and function of full-length and multi-domain constructs of BRCA2 harbouring targeted mutations or deletions. However, as the reviewer highlights, this is technically challenging as existing expression and purification systems for BRCA2 have yields of only ~500 ng from 20 l cultures (<https://www.nature.com/articles/nsmb.1905>), which is insufficient for most structural/biophysical methods, and is impractical for studying multiple separation-of-function mutants. Instead, we believe that the most tractable solution will be to purify the full C-terminal domain of BRCA2 (exons 12-27), corresponding to the globular DNA-binding domain with flanking, MEILB2-binding, Ex14 and Ex27 sequences. Whilst we are setting up a mammalian expression system to produce this BRCA2 construct, this will likely take considerable time to achieve and optimise, so unfortunately remains beyond the scope of the current manuscript. We agree with the reviewer that once such purified constructs are available, their analysis will be truly insightful regarding the coordinated mechanism of action of BRCA2 in recombination. To address this and the point regarding *in vivo* mutants, we have added a paragraph to the discussion in which we address the caveat that findings are based on BRCA2 peptides rather than the full-length protein and summarise the published *in vivo* findings of mouse mutants that affect Ex14 and Ex27 (line numbers 261-276).

Major points:

On the different gels of Figure 1f to h, as well as on Fig 3d, the MBP fusions appear as multiple bands. What are the different species? The authors should give information on that.

2. The additional bands are due to degradation of the fusion proteins to free MBP and/or intermediate species (MBP fused to a partially degraded peptide). This is commonly

observed when expressing short peptides as MBP fusions and the degree of degradation typically depends on the peptide sequence. In this case, degradation was more pronounced for Ex14 and Ex27 than for Ex14-Tr and BRC4. We have added a description of this to the figure legends (line numbers 801-803 and 841-843).

From Figure 1, I am confused with the choice regarding the loops mutant, why not an unrelated loops to show the specificity of the interaction with the DMCA loop?

3. We initially wondered whether swapping the PhePP-binding loops between RAD51 and DMC1 might be sufficient to reverse their binding specificities such that DMC1 would bind to Ex27 (and vice versa for RAD51). Hence, the DMC1 loop mutant consists of the DMC1 sequence with RAD51's PhePP-binding loops. However, whilst the DMC1 loop mutant blocked Ex14-binding, it failed to interact with Ex27. Similarly, the converse mutant of RAD51 harbouring DMC1's PhePP-binding loops also failed to bind to either Ex27 or Ex14. Whilst these mutants demonstrate that the PhePP-binding loops are responsible for their interactions with Ex14 and Ex27, swapping them was insufficient to change the binding specificity. Hence, additional sequence and/or structural differences between DMC1 and RAD51 must contribute to their PhePP-binding specificity. In the original submission, we included the DMC1 loop mutant to confirm the PhePP-binding site, and did not discuss the original aim of changing its specificity. In response to the reviewer's question, we have substantially edited the text to explain the rationale behind the DMC1 loop mutant and how the negative result of it failing to confer Ex27-binding indicates that the divergent PhePP-binding loops are necessary but not sufficient to confer Ex14-/Ex27-binding specificity (line numbers 147-158). Also, we have added in SEC-MALS and SEC-SAXS data that confirm that the octameric ring structure of the DMC1 loop mutant is retained in solution, confirming that the effects on binding are direct rather than being a consequence of a wider structural alteration (Supplementary Figure 6).

We have also added additional data on point mutant V179E of DMC1, which targets the conserved (but divergent from RAD51) valine residue of the PhePP-binding loop that undergoes hydrophobic interactions with amino-acids 2409-PF-2410 of Ex14 (line numbers 158-163 and 194-195). This mutant retains its octameric structure in solution (Supplementary Figure 6) but fails to bind to DMC1 (Figure 2d) or nucleoprotein filaments (Supplementary Figure 9). This mutant directly confirms that PhePP-binding site observed in the crystal structure is responsible for Ex14-binding in solution, without relying on extensive mutants of the loop sequences. Further, it nicely complements our existing data on RAD51 L180E (Figure 4d,e), which is the equivalent mutation of RAD51's PhePP-binding site.

Regarding the stabilization of the filament through Ex14 interaction and the use of the AAA mutant. To me the effect of the AAA mutant is not complete (Figure 2a & 2b for example), suggesting additional effect or missing information. The authors should comment on this or be more precocious with the interpretations.

4. We agree that some stabilisation of DMC1 nucleoprotein filaments is retained for high concentrations of the AAA mutant (shown through partial stimulation of filament formation – Figures 3a,b), although protection is almost completely lost (Figure 2c). We have added EMSA data for the truncated Ex14-Tr peptide, which fully stimulates DMC1-ssDNA filament formation but induces only a slight super-shift and protection from BRC-mediated disruption (Figure 3a-c). This suggests that sequences outside of the PhePP core interaction site, which are present in Ex14 but not Ex14-Tr, may bind across the self-association interface to confer protection. Hence, residual stabilisation by Ex14-AAA may result from this second interaction occurring at high concentration, despite disruption of the PhePP core interaction site. We have commented on this in the manuscript (line numbers 193 and 293-296).

In the different experiments, the implication of BRCA2-Ex14 and BRCA2-Ex27 is shown on preassembled filaments. First, the conditions choose for preassembled (DMC1 11 μ M/ssDNA 33 μ M Fig2a for example) do not appear similar to me on the different gels, band intensities do not look similar? Second, what about the effect of BRCA2 Ex14 on a non-preassembled filament? Is it still possible to detect a stimulation, for example by modifying the kinetic observed, and if no, what is the interpretation? I speculate that this might have implication for the in vivo activity interesting to comment.

5. As the reviewer highlights, the first condition of each experiment should be same for all EMSAs (e.g. panels ii-iv of Figure 3a). We agree that some of the initial conditions had slightly different intensities, although the ratio between bound and free DNA was consistent. This suggests that differences in intensities were due to how the different raw EMSA fluorescence scans were adjusted (brightness/contrast) for presentation rather than being due to underlying differences in starting conditions. In accordance with response number 8, we have reprocessed all EMSAs in the manuscript to ensure visible backgrounds and to provide greater consistency between EMSAs from different scans. In the updated panels (Figures 3a-c and 4e), there is greater consistency in the band intensities of the initial conditions. Further, we have included a Source Data file in which we show the raw scans, initially processed scans (constant brightness adjustment) and final adjusted scans (brightness/contrast) for presentation.

We have repeated the Ex14 stimulation experiment of Figure 3a, changing the order of addition of DMC1, ssDNA and Ex14 (Supplementary 8). We observed the same degree of stimulation of DMC1-ssDNA filament formation and super-shift by Ex14 in all three cases, with no overt differences in the species formed. Hence, Ex14 appears to have the same effect whether the starting species is a pre-assembled DMC1-ssDNA filament or

an Ex14-DMC1 complex (or a mixture of Ex14 and ssDNA). This suggests that the super-shifted species represents a stable Ex14-DMC1-ssDNA that arises through equilibrium rather than through a specifically ordered mechanism. We have added these data to the manuscript (Supplementary 8 and line numbers 186-187 and 819-821).

Regarding protection of RAD51 filament by BRCA2 Ex27 against BRC4 dismantling, is it a total protection? On Figure 3e ii, the effect of BRC4 is still visible. Moreover, I am puzzled by the ratio used for the experiment, how can this huge excess of BRCA2 Ex27 can be interpreted, in particular in a physiological context where the different domains are present on the same protein? Did the authors test different ratios?

6. We agree that the protection shown in Figure 4e is not complete as excess BRC4 was able to partially disrupt filaments. This assay was performed using a random sequence ssDNA substrate as we wanted to use the same substrate for both DMC1 and RAD51 experiments. However, it is known that protection of RAD51-ssDNA filaments by Ex27 is more complete when using polydT rather than random sequence ssDNA substrates (compare Figure panels 6b and 6d of <https://doi.org/10.1038/nsmb1251>). The level of protection conferred to random sequence RAD51-ssDNA filaments shown in 4e is consistent with previous data (Figure 6b of the above reference). In light of the reviewer's comments, we have repeated the RAD51-ssDNA experiments using polydT ssDNA, in which we confirm full protection by Ex27, which is almost entirely lost by the Ex27-AAA mutant or the RAD51 L180E mutant. We have addressed this issue and have added these data to the manuscript (Supplementary Figure 12 and line numbers 236-242).

The choice of a 10-fold molar excess of Ex14/Ex27 (110 μM) in protection assays was to exceed the K_D of the DMC1/RAD51 interaction (which is approximately 30 μM for Ex14-DMC1) in order to favour the PhePP-recombinase interaction, and for consistency with our previous study (<https://doi.org/10.1038/nsmb1251>). Hence, the molar excess is simply required to favour the interaction owing to the micromolar binding affinity. We recognise the need to address this further in the manuscript, so have added discussion of how the concentrations used in the stabilisation, super-shift and protection assays relate to the K_D of the interaction (line numbers 189-192, 201-207 and 242-245). With regards to the physiological context, Ex14 and Ex27 sequences are physically tethered to BRCA2's DNA-binding domain, so would be localised in proximity of nucleating DMC1/RAD51. This would likely have the same effect as increasing the bulk concentration in favouring the interaction.

We have also tested different ratios of Ex14 and Ex27 in protection against a 10-fold excess of BRC4 (Supplementary Figure 10). For Ex14, equimolar quantities conferred some protection (11 μM), whilst a 5-fold excess (55 μM) conferred complete protection. For Ex27, there was no protection at an equimolar amount (11 μM), some

protection at a 3-fold excess (33 μM), and complete protection at a 5-fold excess (55 μM). These findings are consistent with the need for Ex14/Ex27 to be present at a concentration at/above its K_D to promote binding and confer protection from BRC4. We have added these data to the manuscript (Supplementary Figure 10 and line numbers 205-207 and 242-245).

Minor points:

Figure 1e to f: strictly, these experiments do not show that Ex27 interacts with RAD51, which is shown in Fig3, and the literature, the authors should comment on that, or add RAD51 / Ex27 interaction on the gel, to simplify for the reader.

7. Yes, we do not show the RAD51-Ex27 interaction at this stage in the manuscript as it has already been reported in the literature (including <https://doi.org/10.1038/nsmb1251> and <https://doi.org/10.1038/nsmb1245>). We have changed the subheading of this section to refer only to the Ex14-DMC1 interaction (line number 137) and, given the need for additional panels, we have moved previous Figure 1f-i to new Figure 2a-e, which addresses only the Ex14-DMC1 interaction. Hence, we now focus solely on the Ex14-DMC1 interaction at this stage in the manuscript, and then address the Ex27-RAD51 later in the manuscript in Figure 4.

In Figure 3d and e: the gels contrast look very strange to me, with complete blank portion. I am sincerely convinced that the authors did not want to hide any information, however, even of low quality, a gel closer to reality would have been more informative, in particular if residual bands are present, and this should be commented.

8. We agree with the reviewer that the backgrounds of some of the EMSAs and pulldowns were too light, as a consequence of the contrast being too heavily adjusted, and this was more apparent in the pdf rendered version than in the original figure files. We have reprocessed all of the EMSAs and pulldowns in the manuscript to ensure visible backgrounds and to provide greater consistency between EMSAs that were scanned separately (Figures 2b-e, 3a-c and 4d,e). These show no residual bands relative to the original panels. We have also included a Source Data file in which we show the raw scans, initially processed scans (constant brightness adjustment) and final adjusted scans (brightness/contrast) for presentation.

Last section of the result, spelling mistake: BRC2 Ex27 has to be change to BRCA2 Ex27.

9. Thank you for spotting this. We have corrected this error (line number 224).

Reviewer #2 (Remarks to the Author):

The paper by Dunce and Davies reports the crystal structure of the meiotic recombinase DMC1 bound to its specific BRCA2-interaction motif PhePP, together with accompanying biochemical validation of the structure, and modelling of the similar RAD51 - BRCA2 PhePP interaction. The authors find that DMC1 and RAD51 interact in the same way with their respective BRCA2 PhePP motifs, and propose a unified model for how the interaction stabilises the filaments formed by the recombinases on DNA.

This is a nice and timely paper that reports an important advance in our understanding of BRCA2-dependent regulation of meiotic recombination, in agreement with complementary work that has been published very recently (Appleby et al, Nat Comm, 2023) and deposited in bioRxiv (Miron et al, 2023).

I would like to ask the authors to comment on the point below and implement the following minor changes before publication:

The DMC1-binding PhePP motif (FVPPF) is considerably more hydrophobic than the RAD51-binding one, with conservation of a solvent-exposed valine and a second phenylalanine. The additional phenylalanine in particular seems engaged in contacts with DMC1, from Fig. 1c, although the authors do not comment on these. Could this difference help explain the binding preference of DMC1 and RAD51 for their respective PhePP motifs?

10. As the reviewer highlights, the PhePP motif of Ex14 (FVPPFK) is more hydrophobic than Ex27 (FQPPRS). Also, as the reviewer mentions, the V2406 (Ex14) and Q3299 (Ex27) residues are entirely solvent-exposed, so it is difficult to rationalise how they may favour selective DMC1/RAD51-binding. Nevertheless, F2410 (Ex14) and R3302 (Ex27) have different orientations and interactions between Ex14-DMC1 and Ex27-RAD51 structures. F2410 (Ex14) packs against a hydrophobic surface of DMC1 that is formed by its PhePP-binding loops. This interaction may not be possible in RAD51 as this space is occupied by a salt bridge between R150 and the backbone carbonyl of Y178. Conversely, R3302 (Ex27) forms a salt bridge with RAD51 residue D187 (<https://doi.org/10.1038/s41467-023-42830-1>), which would not be possible in DMC1 as its equivalent residue is an alanine. Hence, as the reviewer suggests, differences in hydrophobicity at PhePP residues F2410 and R3302 may contribute towards the DMC1/RAD51-binding specificity of Ex14 and Ex27. Further, the additional hydrophobic contacts of F2410 may enhance the binding affinity such that Ex14-DMC1 binding requires only the core PhePP interaction site whereas Ex27-RAD51 binding depends on a considerably more extensive interface. We agree with the reviewer that this should be discussed in the manuscript. Thus, we extended our description of the PhePP-binding site in the results section (Figure 1d,e and line numbers 127-135), and have added the above comparison of F2410 (Ex14) and R3302 (Ex27) amino-acids to the discussion (line numbers 320-330).

ABSTRACT

The abstract is slightly repetitive and could be improved. Rather than stating once at the start: “BRCA2’s BRC repeats bind and disrupt RAD51 and DMC1 filaments, whereas its PhePP motifs binds to recombinases in a manner that stabilises their nucleoprotein filaments.” and then at the end: “In both cases, BRCA2 PhePP motifs enhance the stability of nucleoprotein filaments, protecting them from BRC-mediated disruption.”, it would be more informative for the reader to replace the second sentence with one explaining how the authors' findings provide insight into the stabilisation of the the DMC1/RAD51 filament by the PhePP motif.

11. We have edited the abstract to avoid the repetition and provide more details of the findings, as suggested (line numbers 25-29).

INTRO

In line 4 of the Introduction, please rephrase “of which the machinery” so that the meaning of the sentence is clearer.

12. We have rephrased this, as requested (line numbers 50-52).

In the introduction, the assertion that isolated RAD51 form large filamentous structure is not quite accurate, please change to ‘heterogeneous oligomers’.

13. We agree with this point and have changed the phrasing as requested (line number 81).

RESULTS

Correct typo: “The PhePP-binding site of DMC1 is formed from by two loops”.

14. Thank you for spotting this. We have corrected this error (line number 127).

Correct typo and clarify meaning of ‘differential PhePP-binding site’ in “The second loop includes amino-acid residues V179 (DMC1) and L180 (RAD51) that substantially contribute to the hydrophobic pocket of the differential PhePP-binding site (Figure 1d).”

15. We have edited the paragraph to clarify the meaning and to provide more details regarding differences in sequences between PhePP-binding loops of DMC1 and RAD51 (line numbers 127-135).

Briefly explain why different buffers represent favourable and unfavourable conditions of filament formations, in “BRCA2 Ex14 stabilises DMC1 nucleoprotein filaments”.

16. The two different buffer conditions are based on ones previously used in the literature to form nucleoprotein filaments. The first condition (TAE) was used for RAD51 (<https://doi.org/10.1038/nsmb1251>), whereas the second (TEA + KCl) was used for DMC1 (<https://doi.org/10.1073/pnas.1920368117>, <https://doi.org/10.1093/nar/gkn352>, <https://doi.org/10.1038/s41467-020-20258-1>). We optimised these and found that the first condition only gave partial filament formation at the canonical binding ratio of one DMC1 protomer to three nucleotides, whereas the latter gave more complete filament formation at the same canonical ratio (compare Figures 3a and 3b). Hence, we referred to the former as unfavourable and the latter as favourable. We chose to include both in the manuscript as the unfavourable conditions was suitable for showing that Ex14 can stimulate full filament formation, whereas the favourable condition was suitable for showing that Ex14 super-shifts and protects preassembled filaments. We have added an explanation of this in the manuscript (line numbers 177-183).

METHODS

I assume that the code in brackets at the end of the subheading refers to the PDB accession code. This might confuse the reader, please remove and provide accession codes under 'data availability'.

17. We have made this change, as requested (line number 434, 453 and 542-550).

Please give residue numbers for the BRCA2 Ex14 peptide used for crystal soaking.

18. The BRCA2 Ex14-Tr peptide (2401-2414) was used for soaking. We have added these details to the methods section (line numbers 457), and have also modified Figure 1a to include the boundaries of all peptides used in the manuscript.

Reviewer #3 (Remarks to the Author):

In this manuscript, Dunce and Davies used crystallography, aided by AlphaFold modeling, to determine the structure of recombinase DMC1 bound to a BRCA2-Ex14 peptide. They show in vitro that binding of BRCA2-Ex14 stimulates DMC1 filament assembly and provides protection against destabilization by the BRCA2-BRC domain. Similarly, RAD51 binding to BRCA2-Ex27 protects against BRC-mediated disruption.

The manuscript provides interesting insights into the role of BRCA2 in mitotic and meiotic recombination. I have a few comments that should be addressed prior to publication.

One limitation of the manuscript is the functional validation of the reported interface. The DMC1 loop mutant involves 12 mutations in total. A more careful residue-by-residue mutational analysis of the interface would be more appropriate. In particular, the importance of DMC1 residue V179 on binding to Ex14 and protection against BRC-mediated disruption should be established.

19. The extensive loop mutation replaces the amino-acids present within the PhePP-binding loops of DMC1 with their RAD51 sequences. The rationale was that we wondered whether this would be sufficient to reverse the binding specificity such that DMC1 would bind to Ex27. This was not the case, as the mutation simply blocked binding to both peptides, indicating that the wider interface contributes to peptide specificity. We have added further details of this to the manuscript (Supplementary Figure 6 and line numbers 147-158), in accordance with our response to a similar point raised in reviewer response number 3.

We have performed the experiment that the reviewer suggests, by introducing a V179E mutation in DMC1, which targets the conserved valine residue of the PhePP-binding loop. This blocked its ability to bind Ex14 (Figure 2d), and underwent only residual super-shift by Ex14 (Supplementary Figure 9). The residual super-shift was the same for Ex14 and Ex14-AAA, indicating that this was likely due to upstream Ex14 sequence binding across the DMC1 self-association interface despite the absence of the anchoring PhePP interface (Supplementary Figure 9). Importantly, we demonstrated that DMC1 V179E remained octameric in solution, confirming that loss of Ex14-binding was a direct consequence of the mutation rather than a structural effect (Supplementary Figure 6). These data confirm that the PhePP-binding pocket mediates the Ex14 interaction without relying on the extensive loop mutation. We have added the V179E data to the manuscript (Figure 2d, Supplementary Figures 6 and 9, and line numbers 158-163 and 194-195).

The authors show that Ex14 does not bind a monomeric DMC1 mutant, and hypothesize that residues upstream the PhePP motif bind across the self-association interface, similarly to the recently reported RAD51-Ex27 interaction (Appleby 2023). However, the crystal structure uses a truncated version of Ex14 that lacks those upstream sequences. The authors should test the prediction that DMC1-F85E binds the Ex14-Tr peptide, and that DMC1 binding to Ex17-Tr does not confer protection against BRC-mediated disruption. Finally, independent analysis of the interaction mode of DMC1-Ex14 (and RAD51-Ex27) across the self association interface, for example through cysteine crosslinking, would be relevant.

20. The reviewer raises an important point that Ex14 selectively binds to oligomeric DMC1 but that the PhePP core interaction site of Ex14-Tr observed in the crystal structure does not include DMC1's self-association interface. As suggested, we have tested whether Ex14-Tr binds to monomeric DMC1 F85E and observed no interaction, similar to Ex14, whereas BRC4 bound to DMC1 F85E in the same manner as wild-type (Figure 2e). Hence, Ex14-Tr selectively binds to oligomeric DMC1 despite interacting with a site that is distant from the self-association interface. The only feasible explanation for this is that specificity is achieved through an allosteric change in DMC1's PhePP-binding site between its oligomeric and monomeric forms, in which binding is enabled only in the former case. We have included the data and a discussion of this in the manuscript (Figure 2e and line numbers 300-307).

We have extended our analysis of stimulation, super-shift and protection of DMC1-ssDNA filaments by Ex14 to include the truncated Ex14-Tr peptide (Figure 3a-c). We found that Ex14-Tr stimulated DMC1-ssDNA filament formation at a level comparable to the full Ex14 peptide (Figure 3a). However, it did not super-shift preassembled filaments (Figure 3b) and largely failed to protect filaments from BRC-mediated disruption (Figure 3c). The failure to super-shift and protect filaments is consistent with our model in which upstream sequences within Ex14 confer protection by binding across the self-association interface, in the same manner as Ex27-RAD51. However, the ability of Ex14-Tr to stimulate filament formation suggests that this function is achieved solely by binding to the PhePP core interaction site. We suggest that this may also be achieved through allostery, in which occupation of the PhePP-binding site favours a conformational change at the self-association interface, such as to favour the formation of ssDNA-bound filaments rather than unbound rings. We have added the data and a discussion of this to the manuscript (Figure 3a-c and line numbers 196-197, 207-209, 280-283 and 307-310).

We agree that it would be ideal for independent analysis to confirm whether the upstream sequence of Ex14 binds across DMC1's self-association interface. We have performed cross-linking mass-spectrometry, aiming to define specific links between Ex14 and DMC1 residues. However, the results were ambiguous as whilst some contacts agreed with our predictions, others were between distant regions of the proteins. As a control, we performed the same cross-linking mass-spectrometry of the BRC4-DMC1 complex, which also gave crosslinks across the DMC1 surface that are

distant from its known binding interface. It is notable that each DMC1 protomer is around 30 Å wide, whereas the BS3 crosslinker length is 10 Å, which including the length of side-chains and some flexibility likely increases the reach to >20 Å. These dimensions are compatible with crosslinks across the majority of the DMC1 surface. Hence, we think that crosslinking-based methods do not have the resolution necessary to either confirm or refute the hypothesis given the scale of this system. Instead, the hypothesised role of upstream Ex14 amino-acids in binding across the DMC1 self-association interface will likely require cryo-EM structure solution of the full Ex14-DMC1-ssDNA complex. We have added a comment to the discussion that upstream binding across the interface must be tested by future structural analysis of Ex14-bound DMC1-ssDNA filaments (line numbers 296-298).

Minor points:

Is the model in Fig SF1 identical to the AlphaFold prediction in SF2? If not, a comparison of the predicted model and experimental structure would be helpful.

21. The peptide conformation in the crystal structure is identical to the AlphaFold2 prediction throughout amino-acids 2406-FVPP-2409, but differs slightly (including between different copies within the crystal) for K2410 and at the C-terminus. We have added a description of this, alongside superpositions of the various copies within the crystal and AlphaFold2 model (Supplementary Figure 3a,b and line numbers 121-125).

A reference to the DMC1-F85E mutant should be provided.

22. This is based on the F86E mutant of RAD51, which disrupts the self-association interface and forms only a monomer (<https://doi.org/10.1038/nature01230>), and was used to show that Ex27 binds oligomeric RAD51 (<https://doi.org/10.1038/nsmb1251>). The F85E mutation is the equivalent mutation of DMC1 that targets its self-association interface in the same way. We have validated this through SEC-MALS analysis of DMC1 F85E, showing that protein only forms monomers (Supplementary Figure 7). We have added the SEC-MALS data and the above description of the mutant to the manuscript (Supplementary Figure 7 and line numbers 165-171).

Typo at the top of page 8: BRC2 Ex27.

23. Thank you for spotting this. We have corrected this error (line number 224).

REVIEWERS' COMMENTS

Reviewer #1 (Remarks to the Author):

The authors have provided a much-improved version of the manuscript, and I am very pleased that they addressed all my concerns, and beyond, by providing a large amount of additional data. In particular, the inclusion of new mutant data (e. g. V179E DMC1) or of pre-assembled filaments look for me of great interest at the mechanistic level.

Overall, the responses and additional work on the manuscript provide a highest level of confidence in the interpretation. I am delighted to recommend that the manuscript is accepted for publication.

Reviewer #2 (Remarks to the Author):

The revisions performed by the authors have addressed my specific points and in general have strengthened an already interesting manuscript.

Reviewer #3 (Remarks to the Author):

The authors have addressed all of my comments. I am happy to recommend publication of this manuscript in its current form and congratulate the authors for their nice work.